# Towards Understanding Generalization of Federated Adversarial Learning: Perspective of Algorithmic Stability

**Yongkang Yang**[1]  **Chang Cao**[1]  **Ke Zhang**[1]  **Han Li**[1]  **Hong Chen**[1]  **Rushi Lan**[2]

## Abstract

Federated Adversarial Learning (FAL) enhances model robustness by integrating adversarial training into the federated learning framework. Despite recent advances proposing efficient FAL algorithms, existing work has mainly focused on convergence properties, with limited understanding of their generalization capabilities. To address this, we present the unified theoretical framework for analyzing FAL generalization through the lens of algorithmic stability. We first analyze general FAL algorithms based on stochastic gradient descent (SGD) and derive perturbation-dependent generalization bounds, which reveal that stronger adversarial attacks can lead to degraded generalization. To mitigate the impact of adversarial perturbations, we leverage Moreau envelope optimization and establish a perturbation-independent bound, demonstrating its efficacy in simultaneously enhancing both robustness and generalization. Finally, we extend our analysis to the practical black-box setting, demonstrating that zeroth-order optimization techniques can effectively maintain both robustness and generalization even without local gradient access.

## 1. Introduction

Federated Learning (FL) is a distributed learning paradigm that enables multiple decentralized clients to collaboratively train models without sharing raw data, ensuring privacy (McMahan et al., 2017; Li et al., 2020; Qu et al., 2022). Despite its privacy advantages, models trained with FL remain vulnerable to adversarial attacks, where imperceptible, adversarially crafted perturbations significantly degrade model performance (Bagdasaryan et al., 2020; Wang et al., 2020; Kumar et al., 2023). To address this issue, recent studies have integrated adversarial training into the FL framework, thereby establishing the paradigm of FAL (Zizzo et al., 2020; Shah et al., 2021). In general, FAL is formulated as a min–max optimization problem, where the global model minimizes the expected loss while client-side adversarial perturbations maximize it, thereby improving robustness against attacks (Zhu et al., 2023; Zhang et al., 2023).

Empirical studies have demonstrated that FAL optimized via SGD is capable of attaining strong robustness on the training data. However, as adversarial training progresses, the robust test accuracy on adversarial examples often declines. This phenomenon is commonly known as robust overfitting (Rice et al., 2020; Yu et al., 2022). Theoretical studies have shown that the non-smooth nature of adversarial loss underlies robust overfitting and the degradation of generalization bounds as attack strength grows (Xing et al., 2021a;b; Xiao et al., 2022). In FAL, the non-smooth inner maximization intrinsically couples the objective's effective regularity with the attack radius $\delta$. This dependency undermines the algorithmic stability of SGD, yielding generalization bounds that unfavorably withand the training rounds $T$. Consequently, this structural characteristic induces a progressive deterioration in robust generalization performance, thereby constituting a fundamental theoretical barrier. Accordingly, Ding et al. (2025) adopted randomized smoothness approximation (Alashqar et al., 2023) to effectively smooth the adversarial loss and applied it to FAL. However, this method relies on a smooth approximation of the non-smooth adversarial loss, retaining an attack-dependent term that requires tighter approximation, thereby increasing computational cost. Moreover, a prevailing limitation of existing studies is their fundamental dependence on first-order gradient information (Xiao et al., 2022; Chen et al., 2022; Zhang et al., 2023). These methodologies necessitate explicit gradient computations via backpropagation to guide the robust optimization process. And in many federated deployments, privacy and system constraints limit clients to black-box access to a prediction service or proprietary model interface, resulting in regimes where gradients are

[1]Engineering Research Center of Intelligent Technology for Agriculture, Ministry of Education, and College of Informatics, Huazhong Agricultural University, Wuhan 430070, China [2]Guangxi Key Laboratory of Image and Graphic Intelligent Processing, Guilin University of Electronic Technology, Guilin 541004, China. Correspondence to: Han Li <lihan125@mail.hzau.edu.cn>.

*Proceedings of the 43rd International Conference on Machine Learning*, Seoul, South Korea. PMLR 306, 2026. Copyright 2026 by the author(s).

*Table 1.* Comparison of stability bounds for Adversarial Learning algorithms ( $n_{\min}$-Minimum local sample size; $D_{\max}$-Maximum heterogeneity measure; $(N/m)$-number of clients; $p$-Moreau parameter; $\ell$-convexity constant; $n$-local sample size; $T$-number of rounds; $\delta$-attack radius; $\mu$-approximation error; $\eta$-step size).

| Algorithm | Federated | Training Mechanism | Analysis Tool | Step Size | Stability Bound |
|---|---|---|---|---|---|
| AT (Xing et al., 2021a) | No | SGD | On-average stability | Constant $(\eta)$ | $\mathcal{O}\left(\frac{\delta\,\eta\sqrt{T}}{n} + \frac{\delta\,\eta T}{n^2}\right)$ |
| AT (Xiao et al., 2022) | No | SGD | Uniform stability | Constant $(1/L_w)$ | $\mathcal{O}\left(\frac{T}{L_w n} + \frac{\delta T}{L_w}\right)$ |
| FAL (Ding et al., 2025) | Yes | SGD + SSA | On-average stability | Diminishing | $\mathcal{O}\left(\delta T \log T\left(1 + \frac{D_{\max}}{m\,n_{\min}}\right) + \frac{T\sqrt{\log T}}{m\,n_{\min}}\right)$ |
| FAL (Ding et al., 2025) | Yes | SGD + RSA | On-average stability | Diminishing | $\mathcal{O}\left(\frac{T^{1/4}\log T}{\sqrt{Q}} + \frac{T^{3/4} + T(\delta D_{\max})^{1/3}}{m\,n_{\min}}\right)$ |
| FAL (Ours) | Yes | SGD | Uniform stability | Diminishing $(\mathcal{O}(1/t))$ | $\mathcal{O}\left(\left(\frac{1}{nN} + \delta\right)T^{1/2}\log T\right)$ |
| FalME (Ours) | Yes | SGD + Moreau | Uniform stability | Diminishing $(\mathcal{O}(1/t))$ | $\mathcal{O}\left(\frac{p}{p-\ell}\left(\frac{1}{Nn} + \frac{1}{n}\right)T^{1/2}\log T\right)$ |
| FalZO (Ours) | Yes | SGD + Zeroth-order | Uniform stability | Diminishing $(\mathcal{O}(1/t))$ | $\mathcal{O}\left(\left(\frac{1}{nN} + \mu + \delta\right)T^{1/2}\log T\right)$ |

unavailable or prohibitively costly (Chen et al., 2017; Ilyas et al., 2018). This obstructs optimization and undermines robust generalization under adversarial perturbations.

To address the above challenges, we systematically analyze the generalization of FAL optimization algorithms via a rigorous stability framework tailored for non-convex settings. Our core contributions are as follows.

- We first demonstrate that standard SGD applied to non-smooth adversarial objectives yields generalization bounds that scale with attack intensity. To mitigate the impact of adversarial perturbations, we incorporate Moreau-envelope smoothing by regularizing the empirical robust loss via a quadratic proximal term. This formulation enables stability bounds independent of the attack strength $\delta$, thereby improving both robustness and generalization.

- Moreover, to adress gradient inaccessibility in black-box deployments, we employ zeroth-order (ZO) adversarial training. This extension eliminates reliance on explicit gradients, thereby broadening applicability to privacy-constrained federated environments. We demonstrate that, although the stronger convergence guarantees of first-order methods, our ZO framework remains theoretically, enabling robust learning solely through black-box queries.

- Extensive experiments on benchmark datasets corroborate our theoretical findings. We show that the Moreau-envelope optimizer significantly narrows the robust-generalization gap and outperforms standard SGD in adversarial accuracy. Furthermore, we demonstrate that our ZO variant maintains competitive robustness and generalization in gradient-free environments.

## 2. Related Work

**Federated Adversarial Learning.** Recently, a sequence of increasingly sophisticated FAL algorithms has emerged.

Zizzo et al. (2020) proposed the FAT framework for federated adversarial training, empirically showing that local PGD adversarial training (Madry et al., 2018) paired with global aggregation improves federated robustness (Zizzo et al., 2020). Subsequently, Shah et al. (2021) introduced FedDynAT, which incorporates dynamic local updates and a regularized aggregation mechanism. This design yields improved natural and adversarial accuracy and faster convergence in their experiments (Shah et al., 2021). To explicitly address aggregation error, Zhou et al. (2022) introduced Fed-BVA, a framework that decomposes server error into bias and variance, utilizing asymmetric adversarial examples to enhance robustness. Building upon the need for efficiency, Hong et al. (2023) developed FedRBN, enabling resource-constrained clients to inherit robustness without intensive local adversarial training. In addition, Zhang et al. (2023) proposed DBFAT, a approach that improves both clean accuracy and adversarial robustness through local re-weighting and global regularization. Furthermore, Zhu et al. (2023) proposed SFAT. It introduces a client-level slack aggregation mechanism and alleviates robustness degradation caused by heterogeneity in locally generated adversarial examples.

**Stability and generalization for adversarial training.** The inherent non-smoothness of adversarial loss functions presents a significant challenge for theoretical analysis. To address it, Xing et al. (2021a) show that residual non-smoothness amplifies algorithmic instability , adding an extra $\mathcal{O}(\eta^2 T)$ term beyond what appears in standard training. Leveraging approximate smoothness, Xiao et al. (2022) and Wang et al. derive stability bounds that scale only linearly with $\eta T$ ($\eta_1$ is initial learning rate) and the perturbation radius (Wang et al., 2024). Then, Cheng et al. (2024) consider generating adversarial examples with a single gradient step and show that this variant admits a tighter algorithmic-stability bound than multi-step adversarial training. In parallel, Xiao et al. (2024) devise new gradient-descent algorithms based on the Moreau envelope to further smooth the loss landscape. Finally, Ding et al. (2025) conduct the first systematic study of robust gener-

alization in heterogeneous FAL across multiple smoothing regimes. Compared with Ding et al. (2025), we establish a unified stability-based theory for both first-order and zeroth-order FAL, and further introduce a Moreau-envelope variant that achieves $\delta$-independent uniform stability, thereby improving upon prior perturbation-dependent guarantees. Table 1 summarizes our main results and compares them with prior stability-based adversarial training algorithms.

## 3. Problem Setting

In a typical FAL framework, $N$ clients collaboratively train a global model by transmitting gradient information to a central server, while keeping their raw data local to preserve privacy. For any $i \in [N]$, $j \in [n]$, let $S := \{S_i\}_{i=1}^N$ represent the collection of all local datasets, where $S_i := \{z_{i,1}, \ldots, z_{i,n}\}$ denote the local dataset, and each sample is independently drawn from the client-specific distribution $\mathcal{D}_i$. We denote $w \in \mathcal{W} \subseteq \mathbb{R}^d$ as the global model parameters and $w_i$ as the local model parameters of each client. The standard (non-adversarial) loss function for client $i$ is defined as $g_i(w_i; z_{i,j})$, which measures the discrepancy between the model prediction parameterized by $w_i$ and the observed data sample $z_{i,j}$. Unless otherwise stated, we impose no convexity assumption on the loss.

In the FAL framework, we consider a surrogate adversarial loss that evaluates the model's performance under worst-case perturbations. For any client, the local adversarial loss $f_i(w_i; z_{i,j})$ is formulated as the solution to an inner maximization problem:

$$f_i(w_i; z_{i,j}) = \max_{\|z_{i,j} - z'_{i,j}\|_\infty \leq \delta} g_i(w_i; z'_{i,j}), \qquad (1)$$

where $\delta > 0$ specifies the $\ell_\infty$-norm bound on admissible perturbations.

The objective of FAL is to minimize the global population risk by collaboratively optimizing model parameters across all $N$ clients. Specifically, the population risk for the $i$-th client is defined as:

$$F_i(w_i) = \mathbb{E}_{z_i \sim \mathcal{D}_i}[f_i(w_i; z_i)], \qquad (2)$$

let $z_i$ be a random variable following the local data distribution $\mathcal{D}_i$ associated with the $i$-th client, where $\mathbb{E}_{z_i \sim \mathcal{D}_i}[\cdot]$ denotes the expectation with respect to the sampling of $z_i$. Then the corresponding global population is given by

$$F(w) = \frac{1}{N} \sum_{i=1}^N F_i(w_i) = \frac{1}{N} \sum_{i=1}^N \mathbb{E}_{z_i \sim \mathcal{D}_i}[f_i(w_i; z_i)]. \quad (3)$$

As the population risk cannot be evaluated exactly in practice, we approximate it with the empirical risk computed over the finite dataset $S_i$:

$$F_{S_i}(w_i) = \frac{1}{n} \sum_{j=1}^n f_i(w_i; z_{i,j}), \qquad (4)$$

and the global empirical risk is given by

$$F_S(w) = \frac{1}{N} \sum_{i=1}^N F_{S_i}(w_i) = \frac{1}{N} \sum_{i=1}^N \frac{1}{n} \sum_{j=1}^n f_i(w_i; z_{i,j}),$$

where $z_{i,j}$ represents the $j$-th sample of the $i$-th client, $F_{S_i}(w_i)$ and $F_S(w)$ denote the empirical risks for the local and global models.

### Stability and Generalization

With the adversarial objective defined above, we now turn to the analysis of FAL. In this section, we analyze the generalization properties of FL models under adversarial settings using stability-based methods. Let $A$ be a randomized FAL algorithm. Given a training dataset $S$, we define $w^T = A(S)$ as the global model parameters returned by $A$ after $T$ iterations. To evaluate the generalization performance of the learned model, we define the expected adversarial generalization error as the discrepancy between its population risk and empirical risk:

$$\mathcal{E}_{gen} = \mathbb{E}_{S,A}\left[F(A(S)) - F_S(A(S))\right], \qquad (5)$$

where the expectation is taken over the random sampling of the dataset $S$ and the potential randomness of algorithm $A$.

In FAL, the expected generalization risk can be decomposed as:

$$\mathcal{E}_{gen} = \mathbb{E}_{S,A}\left[\frac{1}{N} \sum_{i=1}^N \left(F_i(A(S)) - F_{S_i}(A(S))\right)\right], \quad (6)$$

which reflects that the generalization ability of the global server model depends on the contributions from all clients. Here, $A(S)$ is the global model returned after $T$ iterations of any FAL algorithm $A$ on $S$. We consider three instantiations: Algorithm 1 (FAL), Algorithm 2 (FalZO), and Algorithm 3 (FalME), with details in Appendix D.

We assume that the loss function $g_i$ of the $i$-th client satisfies the following assumption:

**Assumption 3.1.** Assume that the loss function $g_i(w_i, z_i)$ satisfies the following Lipschitz conditions, where $w_i$ denotes the parameter of the local model and $z_i$ is the input data sample.

$(a)$ (Lipschitz continuity regarding $w_i$ ). For any $w_i, \bar{w}_i \in \mathcal{W}$, $i \in [N]$, and $L > 0$, we have

$$\|g_i(w_i, z_i) - g_i(\bar{w}_i, z_i)\| \leq L\|(w_i - \bar{w}_i)\|.$$

(b) (Lipschitz smoothness regarding $w_i$). For any $w_i, \bar{w}_i \in \mathcal{W}, i \in [N]$, and $L_w > 0$, $g_i$ satisfies

$$\|\nabla_{w_i} g_i(w_i, z_i) - \nabla_{w_i} g_i(\bar{w}_i, z_i)\| \le L_w \|w_i - \bar{w}_i\|.$$

(c) ( Lipschitz smoothness regarding $z_i$). For any $w_i \in \mathcal{W}$, $i \in [N]$, and $L_z > 0$, we have

$$\|\nabla_{w_i} g_i(w_i, z_i) - \nabla_{w_i} g_i(w_i, z_i')\| \le L_z \|z_i - z_i'\|.$$

Note that Assumption 3.1 is widely used in existing research (Xing et al., 2021a; Kanai et al., 2023; Ding et al., 2025), and it is satisfied by widely used losses such as squared loss and cross-entropy.

Inspired by (Hardt et al., 2016), we analyze the generalization behavior of FAL through the lens of algorithmic stability. In particular, we extend the notion of $\epsilon$-uniform stability to the federated setting by characterizing stability with respect to each client-level objective $f_i$.

**Definition 3.2.** A randomized FAL algorithm $A$ is said to be $\epsilon$-uniformly stable if, for all datasets $S$ and $S'$ that differ in at most one sample (possibly belonging to any client), the following holds.

$$\sup_{(i,z)} \mathbb{E}_A \left[ f_i(A(S_i); z) - f_i(A(S_i'); z) \right] \le \epsilon,$$

where the expectation is taken over the internal randomness of $A$, and $f_i(\cdot; z)$ denotes the adversarial loss on client $i$ evaluated at an arbitrary test example $z \sim \mathcal{D}_i$.

**Theorem 3.3.** *Suppose a randomized FAL algorithm $A$ is $\epsilon$-uniformly stable. Then, the expected generalization gap satisfies*

$$\left| \mathbb{E}_{S,A} \left[ \frac{1}{N} \sum_{i=1}^{N} \left( F_i(A(S)) - F_{S_i}(A(S)) \right) \right] \right| \le \epsilon. \quad (7)$$

Theorem 3.3 provides a formal connection between stability and generalization. It shows that the expected generalization error is controlled by the uniform algorithmic stability $\epsilon$. Distinct from centralized paradigms, this federated bound is characterized as an average of client-level discrepancies, revealing that the sensitivity to individual data points is doubly attenuated by the interplay of the local sample size $n$ and the global aggregation factor $1/N$.

## 4. Generalization Analysis

In this section, we introduce our proposed FAL algorithms and derive rigorous generalization guarantees within the uniform stability framework. Our analysis operates exclusively within the general non-convex regime, addressing the practical complexities of deep learning objectives.

### 4.1. SGD-based Optimization

We begin with the classical SGD-based instantiation of FAL, which integrates adversarial training into the Federated Averaging (FedAvg) framework. In each communication round, a subset of clients performs local adversarial training for $K$ epochs and uploads the updated parameters to the server. Subsequently, the server aggregates these updates via averaging. As shown below

$$w^{t+1} = \frac{1}{m} \sum_{i=1}^{m} w_i^{t+1}, \quad (8)$$

where $w_i^{t+1}$ denotes the local parameter after $K$ local epochs on client $i$. Equivalently, viewing the whole procedure as a stochastic optimization process with adversarial gradients, the local update admits the following SGD form:

$$w_i^{t+1} = w_i^t - \frac{\eta_t}{b_1 M} \sum_{i \in \mathcal{M}_t} \sum_{j \in [n]} \alpha_{i,j}^t \nabla f_i(w_i^t; z_{i,j}^t), \quad (9)$$

here, $\eta_t = \eta_1 t^{-1}$ represents the step size at iteration $t \in [T]$, where $\eta_1$ denotes the initial learning rate. The subset of $M$ clients selected at each iteration is denoted by $\mathcal{M}_t \subseteq [N]$ ($|\mathcal{M}_t| = M$), and $\alpha_{i,j}^t$ signifies the count of local samples identical to $z_{i,j}^t$.

To establish theoretical generalization bounds for FAL, we first ground our analysis in the framework of algorithmic stability. Specifically, we bound the expected parameter divergence between models trained on a dataset $S$ and a neighboring dataset $S'$ differing by a single sample.

**Theorem 4.1.** *Let $\{w^t\}$ and $\{\bar{w}^t\}$ be the parameter sequences produced by the FAL algorithm on the dataset $S$ and its neighboring dataset $S'$. With a diminishing step size $\eta_t = \eta_1 t^{-1}, \eta_1 \le \frac{1}{2L_w}$, the final output of the algorithm after $T$ iterations is denoted by $w^T$. Then, we have:*

$$\frac{1}{nN} \sum_{i=1}^{N} \sum_{j=1}^{n} \mathbb{E} \left[ \|w^T - \bar{w}^T\| \right]$$

$$\le (e(T-1))^{\eta_1 L_w} \left( \frac{2L}{nN} + 2L_z \delta \right) \eta_1 \log(e(T-1)).$$

*Remark* 4.2. The stability bound established in Theorem 4.1 explicitly characterizes the dependence of model sensitivity on the adversarial perturbation $\delta$. Crucially, the term $L_z \delta$ indicates that stronger adversarial attacks introduce additional instability. By leveraging the connection between uniform stability and generalization error, we derive the following bound on the expected generalization gap.

**Theorem 4.3.** *For the FAL algorithm $A$. With a diminishing step size $\eta_t = \eta_1 t^{-1}, \eta_1 \le \frac{1}{2L_w}$, and under Assumption 3.1, the expected generalization gap satisfies:*

$$\mathcal{E}_{gen} \le \mathcal{O} \left( L \left( (nN)^{-1} L + L_z \delta \right) T^{\frac{1}{2}} \log T \right).$$

*Remark* 4.4. Under the non-smooth adversarial objective, the induced generalization bound scales as $\mathcal{O}(T^{1/2} \log T)$ with respect to communication rounds. Importantly, the bound exhibits a linear dependence on the adversarial attack strength $\delta$ and the landscape smoothness parameter $L, L_z$. Unlike the standard generalization error term which scales with the inverse sample size $(nN)^{-1}$. Consequently, stronger attacks and rougher loss landscapes inherently exacerbate the generalization gap, theoretically validating the robust overfitting phenomenon and underscoring the necessity for federated methods that effectively regularize the training process to control instability.

## 4.2. Moreau Envelope-based Optimization

In this section, we generalize the FAL framework by integrating Moreau-envelope smoothing to mitigate the intractability caused by the inherent non-smoothness of adversarial objectives. We begin by characterizing a key structural property of the adversarial loss. Although it is non-smooth, it remains weakly convex due to the smoothness of the underlying loss function. Folling (Nurminski, 1973), we formally define this structural property, as follows:

**Definition 4.5.** Let $l > 0$. A function $f$ is said to be $l$-weakly convex if for all $x$, the function $f(x) + \frac{l}{2}\|x\|^2$ is convex in $x$.

Based on this property, we construct a differentiable surrogate for client $i$ by augmenting the empirical robust loss with a quadratic proximal term controlled by a smoothing parameter $p > l$. The empirical Moreau envelope objective is formally defined as:

$$M_i(u_i; S_i) = \min_{w_i} \left\{ \frac{1}{n} \sum_{j=1}^{n} f_i(w_i; z_{i,j}) + \frac{p}{2}\|w_i - u_i\|^2 \right\}.$$

We employ an alternating minimization strategy wherein the inner proximal objective, rendered strictly strongly convex under the condition $p > l$, admits a unique global minimizer $w_i^*$. Subsequently, the outer variable $u_i$ is updated via gradient descent on the smooth envelope. The gradient admits the closed-form expression $\nabla_{u_i} M_i(u_i; S_i) = p(u_i - w_i^*)$. Consequently, the local update rule at iteration $t$ is formulated as:

$$u_i^{t+1} = u_i^t - \alpha_t \nabla_{u_i} M_i(u_i^t; S_i) = u_i^t - \alpha_t p(u_i^t - w_i^*). \quad (10)$$

where $\alpha_t$ denotes the step size and $w_i^*$ represents the optimal solution to the inner minimization at step $t$.

To establish theoretical generalization bounds for FAlME, we first ground our analysis in the framework of algorithmic stability.

**Theorem 4.6.** *Let $\{u_i^t\}$ and $\{\bar{u}_i^t\}$ denote the parameter sequences generated by the FalME algorithm on the dataset*

$S$ *and its neighboring dataset $S'$, respectively. Suppose the loss function is $l$-weakly convex, satisfying Assumption 3.1. With a smoothing parameter $p > l$ and a diminishing step size $\alpha_t$, let $u^T$ be the final global model parameters after $T$ iterations. Then, we have:*

$$\mathbb{E}\left[\|u^T - \bar{u}^T\|\right]$$
$$\leq \mathcal{O}\left(\left(\frac{Lp}{Nn(p-l)} + \frac{p}{n(p-l)}\right) T^{\alpha_1 L_w} \log T\right),$$

*where $L_w = \max\{p, \frac{pl}{p-l}\}$ denotes the smoothness constant under the Assumption 3.1.*

*Remark* 4.7. Note that, the denominator term $(p - l)$ characterizes an inherent approximation-stability trade-off, indicating that tighter smoothing approximations (as $p \to \ell$) exacerbate optimization instability. Utilizing this uniform stability guarantee, we establish the following bound on the expected generalization gap.

**Theorem 4.8.** *Assume that each loss function $f_i$ be $L$-Lipschitz continuous and $l$-weakly convex. If we execute the FalME algorithm with a smoothing parameter $p > l$ and a diminishing stepsize $\alpha_t = \alpha_1/t$ satisfying $\alpha_1 < 1/L_w$, $L_w = \max\{p, \frac{pl}{p-l}\}$. Then the expected generalization gap satisfies*

$$\mathcal{E}_{gen} \leq \mathcal{O}\left(\frac{Lp}{p-l}\left(\frac{L}{Nn} + \frac{1}{n}\right) T^{\frac{1}{2}} \log T\right),$$

*where the initial stepsize $\alpha_1 = \frac{1}{2L_w}$.*

*Remark* 4.9. Theorem 4.8 establishes the theoretical generalization guarantee for FalME, explicitly characterizing how the generalization gap scales with the smoothing parameter $p$, the effective sample size $Nn$, and the training rounds $T$. Specifically, the condition $p > l$ serves a dual purpose: it acts as a geometric regularizer to induce strong convexity in the inner subproblem while determining the effective smoothness $L_w$ of the outer objective. Unlike standard nonconvex approaches that suffer from exponential instability, FalME recovers the canonical rate $\mathcal{O}(\sqrt{T})$ typical of convex optimization, provided the inner solver is sufficiently accurate. Furthermore, the dependence on $(Nn)^{-1}$ confirms the theoretical advantage of federated averaging: the generalization gap is suppressed linearly by the total effective sample size. For comparison, directly applying SGD to the federated adversarial setting yields a generalization bound of $\mathcal{E}_{gen} \leq \mathcal{O}\left(L((nN)^{-1}L + L_z\delta)T^{\frac{1}{2}} \log T\right)$. Importantly, the baseline bound is hindered by the perturbation term $L_z\delta$, which results in an irreducible asymptotic error floor. In contrast, FalME circumvents this limitation, ensuring the generalization gap strictly vanishes with a sufficiently large sample size.

## 4.3. Zeroth-Order Optimization

Beyond first-order methods, we consider scenarios where gradient information is unavailable or inaccessible. Zeroth-

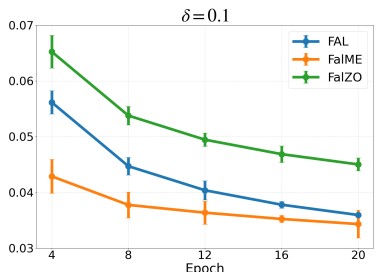 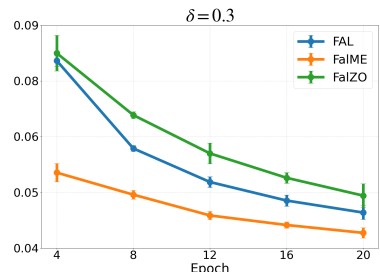 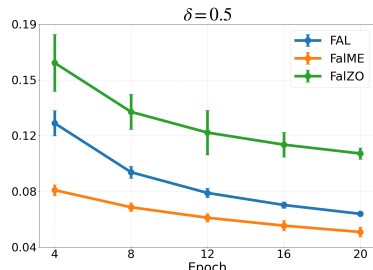

*Figure 1.* Results for generalization gap with different of attack strength ($\delta$) for different algorithms on MNIST.

order optimization (ZOO) trains models by estimating gradients using only function evaluations, thus eliminating the need for explicit derivative information (Chen et al., 2017; Nikolakakis et al., 2022). Building upon this approach, the federated zeroth-order framework FedZO (Fang et al., 2022; Chen et al., 2023), along with its adversarial variant FalZO, extends gradient-free learning to federated settings. In this framework, each client performs private or black-box optimization using only function queries, without requiring access to model gradients.

In each communication round of FalZO, the optimization process is driven by the aggregation of local zeroth-order updates derived from a selected subset of clients. Formally, let $\mathcal{M}_t$ (with $|\mathcal{M}_t| = M$) denote the set of selected clients in round $t$, and $\eta_t$ represent the learning rate. The global model update rule is governed by:

$$w^{t+1} = w^t - \frac{\eta_t}{b_1 M} \sum_{i \in \mathcal{M}_t} \sum_{m=1}^{b_1} \tilde{\nabla} f_i \left( w_i^t; z_{i,m}^t, \{v_{il}^t\}_{l=1}^{b_2}, \mu \right),$$

where $b_1$ is the local batch size, and $\tilde{\nabla} f_i$ represents the zeroth-order gradient estimator. To circumvent the need for backpropagation, FalZO approximates this gradient term via a randomized finite-difference scheme using only function evaluations. Specifically, for a local model parameter $w_i^t$, a data sample $z_{i,m}^t$, and a smoothing parameter $\mu$, the estimator is computed over a set of $b_2$ random direction vectors $\{v_{il}^t\}_{l=1}^{b_2}$ as:

$$\tilde{\nabla} f_i \left( w_i^t; z_{i,m}^t, \{v_{il}^t\}_{l=1}^{b_2}, \mu \right)$$
$$= \frac{1}{b_2} \sum_{l=1}^{b_2} \frac{v_{il}^t}{\mu} \left( f_i \left( w_i^t + \mu v_{il}^t; z_{i,m}^t \right) - f_i \left( w_i^t; z_{i,m}^t \right) \right),$$

where $v_{il}^t$ are independent random unit vectors. To guarantee a high-fidelity approximation of the true gradient, the number of random directions is typically chosen to satisfy $b_2 > d$, where $d$ corresponds to the dimension of the model parameters (Nikolakakis et al., 2022).

Proceeding analogously to the analysis of FAL, we first establish the algorithmic stability for FalZO. By bounding

the parameter divergence between the algorithm running on adjacent datasets $S$ and $S'$, we obtain the following stability guarantee.

**Theorem 4.10.** *Let $\{w^t\}$ and $\{\bar{w}^t\}$ be produced by the FalZO algorithm on the dataset $S$ and its neighboring dataset $S'$. Under a diminishing step size $\eta_t = \eta_1 t^{-1}$, let $w^T$ denote the final output after $T$ iterations. Specifically, for the FalZO variant, we define the coefficient $a_1 = (1 + \sqrt{d/b_2})L_w, \eta_1 \leq \frac{1}{2a_1}$. Then, the following holds:*

$$\frac{1}{nN} \sum_{i=1}^{N} \sum_{j=1}^{n} \mathbb{E} \left[ \|w^T - \bar{w}^T\| \right]$$
$$\leq (e(T-1))^{a_1 \eta_1} a_2 \eta_1 \log(e(T-1)),$$

*where $a_2$ is given by*

$$\left( \frac{2L}{nN} + \mu L_w + 2L_z \delta + \frac{2ML_z \delta}{N} + (2L_z \delta + \frac{2L}{nN})\sqrt{\frac{d}{b_2}} \right).$$

Leveraging this stability guarantee, we derive the generalization bound for FalZO, which explicitly accounts for the additional bias introduced by ZO gradient estimation.

**Theorem 4.11.** *Under Assumption 3.1, suppose the FalZO algorithm is executed with diminishing stepsizes $\eta_t = \eta_1/t, \eta_1 \leq \frac{1}{2a_1}$. Then, the expected generalization gap satisfies:*

$$\mathcal{E}_{gen} \leq \mathcal{O} \left( L \left( (nN)^{-1} L + \mu + L_z \delta) \right) T^{\frac{1}{2}} \log T \right).$$

*Remark* 4.12. Based on the diminishing stepsize schedule $\eta_t = \eta_1/t$ and the smoothnes condition specified in Assumption 3.1, the algorithm FalZO enjoys a data-dependent generalization bound that scales polynomially with the total sample size $nN$ and depends on the zeroth-order approximation error $\mu$ and the perturbation factor $L_z \delta$. Notably, when the smoothing parameter is set to $\mu = \mathcal{O}\left((nN)^{-1}\right)$, the bound tightens to the order of $\mathcal{O}\left(L((nN)^{-1}L + L_z\delta)T^{\frac{1}{2}}\log T\right)$. Moreover, if the problem parameters satisfy $\frac{L_w c}{L_w c + 1} \geq \frac{1}{2}$ for some constant $c > 0$, this canonical rate remains attainable.

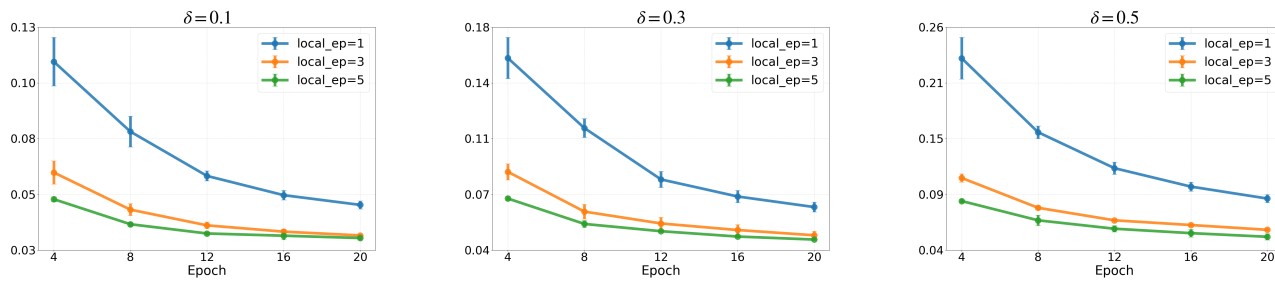

*Figure 2.* Results for generalization gap with different local training epochs on MNIST

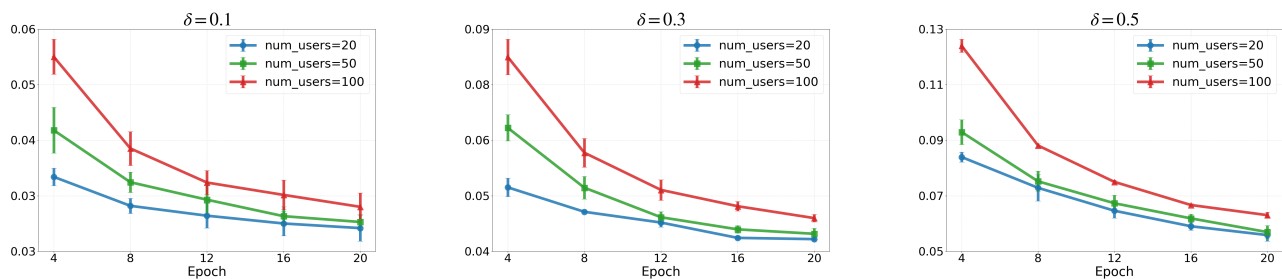

*Figure 3.* Results for generalization gap with different total client numbers on MNIST

## 5. Experiments

### 5.1. Experimental Setup

**Datasets.** We conduct extensive experiments using real-world datasets from public repositories, comprising standard image benchmarks (MNIST, CIFAR10, SVHN) and large-scale vector datasets (Webspam, Epsilon, SUSY) sourced from the LIBSVM library. Statistics of the datasets are provided in Table 2.

**Model Settings.** We adopt a convolutional neural network (CNN) with two convolutional layers (10 and 20 filters, kernel size 5), each followed by ReLU activation, max pooling, and dropout. Features are fed into a fully connected layer with 50 units and dropout, and a final output layer for classification. All models are trained with an initial learning rate of $0.01$ (decayed during training), batch size $10$, and local updates are performed for 5 epochs on each client per communication round. The MNIST dataset is partitioned among $100$ clients in an IID manner, with $50\%$ of clients randomly selected to participate in each round.

Adversarial robustness is assessed via the $\ell_\infty$-PGD attack configured with a budget $\delta$ (10 iterations, step size $\delta/5$). For optimization, the algorithm adapts to gradient accessibility: we employ first-order SGD (momentum 0.5) when gradients are available, and resort to zeroth-order methods in black-box settings. In the latter case, gradient estimation relies on random direction finite differences, coupled with the Moreau envelope to effectively smooth the non-differentiable adversarial objective.

**Evaluation Metric.** We report the adversarial generalization gap, defined as the absolute difference between adversarial training accuracy and adversarial test accuracy.

$$\text{Generalization Gap} = |\text{AdvTrainAcc} - \text{AdvTestAcc}|.$$

*Table 2.* The details of the adopted datasets.

| Dataset | Size ($n$) | Dimension ($d$) |
|---|---|---|
| MNIST | 70,000 | 2784 |
| CIFAR10 | 60,000 | 3072 |
| SVHN | 99,289 | 3072 |
| Webspam | 350,000 | 254 |
| Epsilon | 400,000 | 2,000 |
| SUSY | 5,000,000 | 18 |

### 5.2. Experimental analysis

**Impact on different algorithm.** As illustrated in Figure 1, the detrimental effect of adversarial perturbations varies across the evaluated methods. Specifically, FalME demonstrates the highest resilience, exhibiting the mildest performance degradation, followed by the classical FAL, while FalZO shows the most sensitivity. This suggests that Moreau-envelope smoothing serves as a robust defense mechanism against adversarial attacks. Meanwhile, although FalZO remains a viable gradient-free alternative, it is inherently more susceptible to strong perturbations. Furthermore, Figure 5 compares the average communication

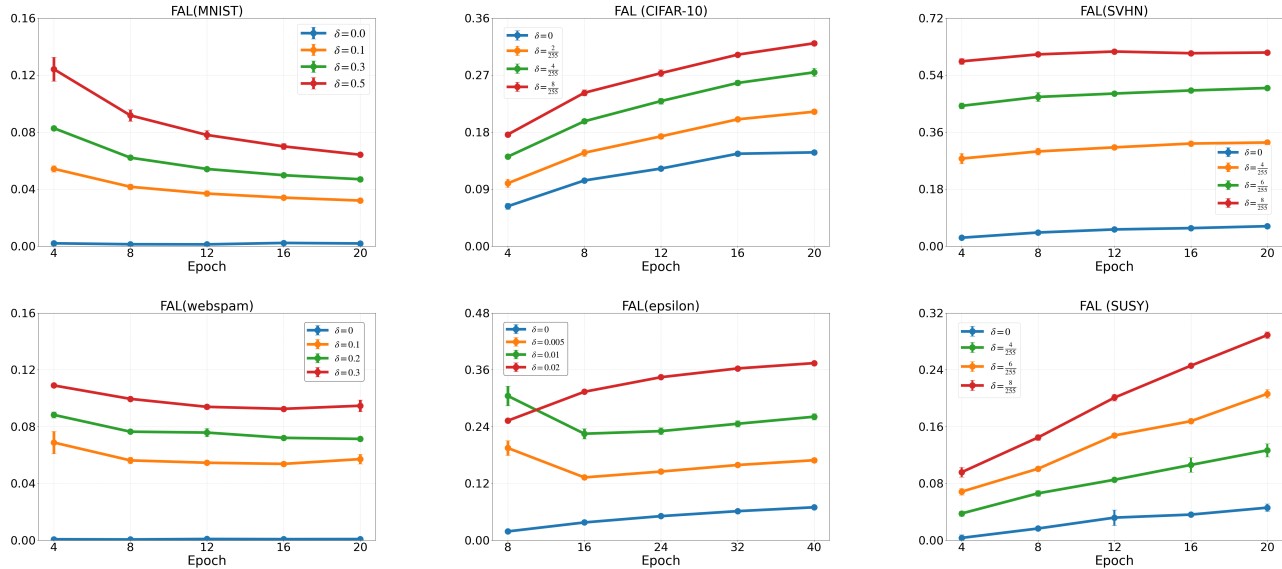

*Figure 4.* The generalization gap with different attack strengths ($\delta$) across different datasets.

efficiency of FalME and FAL under identical experimental conditions. The results indicate that FalME achieves this superior robustness without incurring significant computational or communication overhead, maintaining runtime efficiency comparable to that of FAL.

**Impact on $\delta$.** Extensive experiments across three image datasets (MNIST, CIFAR-10, SVHN) and three large-scale vector datasets in Figure 4 demonstrate a consistent trend: increasing the attack radius $\delta$ universally exacerbates the generalization gap. This confirms that stronger adversarial perturbations inherently degrade the model's ability to generalize, particularly hindering convergence in the early training stages.

**Impact on local training epochs.** As illustrated in Figure 2, increasing the number of local iterations effectively accelerates optimization and significantly narrows the generalization gap, particularly in the early stages. This deeper local optimization enhances both communication efficiency and model robustness. Although excessive updates may risk overfitting, a moderately increased iteration budget consistently improves global generalization performance.

**Impact of total client numbers.** Under a fixed global data budget, increasing the number of clients $N$ leads to data fragmentation and local sample scarcity. As illustrated in Figure 3, this reduction in local data density correlates with a marked expansion of the generalization gap. This highlights that excessive data partitioning compromises the statistical representativeness of local updates, thereby exacerbating local overfitting and hindering robust learning.

**Impact on client participation ratio.** Increasing the client participation ratio incorporates a larger portion of the dataset in each communication round, thereby facilitating the training of more accurate global models. This leads to a reduction in the generalization gap and an improvement in robust accuracy, as illustrated in Appendix D, Figure 6.

## 6. Conclusion

In this paper, we develop a unified theoretical framework for Federated Adversarial Learning that covers both first-order and zeroth-order optimization. We show that classical smooth-surrogate methods suffer from a perturbation-dependent bias, causing generalization to degrade as attack strength increases. To address this, we propose FalME, a first-order method based on Moreau-envelope smoothing that separates non-smoothness from optimization, yielding perturbation-independent generalization bounds while recovering the canonical convergence rate. Subsequently, we further extend the analysis to the zeroth-order setting via FalZO, establishing robust generalization in privacy-sensitive black-box regimes. Extensive experiments support our theory, demonstrating mitigation of robust overfitting and offering a practical reference for scalable distributed adversarial learning.

## Impact Statement

This paper presents work whose goal is to advance the field of machine learning. There are many potential societal consequences of our work, none of which we feel must be specifically highlighted here.

## Acknowledgments

This work was supported in part by the National Natural Science Foundation of China (NSFC) under Grant No. 62376104 and the R&D Program under Grant No. AB25069496.

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

# A. Notations

The main notations of this paper are summarized in Table 3.

*Table 3.* Summary of main notations involved in this paper

| Notations | Descriptions |
|---|---|
| $N, n$ | the total number of clients and the total sample number of each client |
| $\mathcal{M}_t, M$ | the set of client indices selected at the $t$-th iteration, and the number of clients in this set |
| $S, S_i$ | the total dataset and the local dataset of the $i$-th client, where $i = 1, \ldots, N$ |
| $z_{ij}$ | the $j$-th sample of $\mathcal{S}_i$ sampled from $\mathcal{D}_i$, $j = 1, \ldots, n$ |
| $\mathcal{D}_i$ | the distribution of $z_{ij}$ with $\mathcal{D}_i$ independent across clients, i.e., $\mathcal{D}_i$ is independent of $\mathcal{D}_{i'}$ |
| $w_t, w_i^t$ | the parameters $w^t$ for the global model and $w_i^t$ for the $i$-th local model at iteration $t$ |
| $F(w)$ | the expected risks for the global mode |
| $F_i(w_i)$ | the expected risks for the local model of the $i$-th client |
| $F_{S_i}(w)$ | the empirical risks for the local model of the $i$-th client |
| $F_S(w)$ | the empirical risks for the global model |
| $f_i(w_i, z_i), \nabla f_i, \widetilde{\nabla} f_i$ | the loss function of the $i$-th client, its gradient, and the estimated gradient, respectively |
| $\delta$ | the magnitude of the adversarial perturbation |
| $\epsilon$ | the parameter of $\ell_1$ on-average model stability |
| $T$ | the total number of iterations |
| $\mathcal{A}^S$ | the random algorithm based on dataset $S$ |
| $A, A(S) = w^T$ | the federated adversarial learning algorithm and the parameter trained with A on S |
| $L, L_\theta$ | the parameters of Lipschitz, smoothness respectively |
| $b_1, b_2$ | the sizes of i.i.d. random samples and random direction vectors |
| $\eta_t, \mu$ | the step size $\eta_t$ at the $t$-th iteration and the positive increment $h$ in the derivative |
| $\epsilon_{\text{gen}}$ | the generalization error in the adversarial settings |
| $v_{il}^t$ | the $l$-th random direction vector for the $i$-th client in the $t$-th iteration |
| $p$ | the positive regularization parameter in the Moreau envelope |
| $u$ | the auxiliary/global model parameter in the outer minimization |
| $\alpha$ | the step size for the outer update (global Moreau envelope gradient step) |
| $K_i(w_i, u_i; z_i)$ | the Moreau envelope-regularized loss for client $i$ |
| $M_i(u_i; S_i)$ | the Moreau envelope of empirical loss for client $i$ |
| $M(u; S)$ | the global Moreau envelope loss over all clients |
| $w_i(u_i; S_i)$ | the local minimizer for the Moreau-regularized loss on client $i$ |

# B. Some Useful Lemmas

We first introduce the lemmas which will be used in our proofs.

**Lemma 1.1.** *Let $f_i$ be the adversarial loss defined in (1) and $g_i$ satisfies Assumption 3.1. $\forall w_i^t, \bar{w}_i^t$ and $\forall z_i \in Z$, the following properties hold.*

$$\|\nabla F_S(w^{(1)}) - \nabla F_S(w^{(2)})\| \le L_w \|w^{(1)} - w^{(2)}\| + 2L_z\delta,$$

where $L_w$ represents the smoothness parameter.

**Lemma 1.2** (Li & Liu, 2022)**.** *Let $e$ be the base of the natural logarithm. The following inequalities hold:*

*(a) if $\alpha = 1$, then $\sum_{k=1}^{t} k^{-\alpha} \le \log(et)$; (b) if $\alpha > 1$, then $\sum_{k=1}^{t} k^{-\alpha} \le \dfrac{\alpha}{\alpha - 1}$.*

**Lemma 1.3** (Duchi et al., 2015)**.** *Assme a random vector $X \in \mathbb{R}^d$ is $d$-dimensional uniform distribution. For any $k \in \mathbb{N}$, there holds $\mathbb{E}[\|X\|^k] = d/(d+k)$.*

**Lemma 1.4** (Duchi et al., 2015)**.** *Let $v_l \in \mathbb{R}^d$, $l \in \{1, 2, \ldots, b_2\}$ be i.i.d. random vectors satisfying $d$-dimensional uniform*

*distribution. For every random vector $u \in \mathbb{R}^d$ independent of all $v_l$, the following inequality holds*

$$\mathbb{E}\left[\left\|\frac{1}{b_2}\sum_{l=1}^{b_2}\langle u, v_l\rangle v_l - u\right\| \Big| u\right] \le \sqrt{\frac{d}{b_2}}\|u\|.$$

# C. Proofs of Main Results

## C.1. Proof of Lemma 1.1.

According to the definition of global empirical risk, and from (Xiao et al., 2022) Lemma 4.1, consider parameters $w^{(1)}, w^{(2)}$, the gradient difference is bounded by

$$\|\nabla F_S(w^{(1)}) - \nabla F_S(w^{(2)})\|$$
$$= \left\|\frac{1}{N}\sum_{i=1}^{N}\frac{1}{n}\sum_{j=1}^{n}[d_i(w_i^{(1)}, z_{i,j}) - d_i(w_i^{(2)}, z_{i,j})]\right\|$$
$$\le \frac{1}{N}\sum_{i=1}^{N}\frac{1}{n}\sum_{j=1}^{n}\|d_i(w_i^{(1)}, z_{i,j}) - d_i(w_i^{(2)}, z_{i,j})\|$$
$$\le \frac{1}{N}\sum_{i=1}^{N}\frac{1}{n}\sum_{j=1}^{n}[L_w\|w_i^{(1)} - w_i^{(2)}\| + 2L_z\delta]$$
$$= L_w\|w^{(1)} - w^{(2)}\| + 2L_z\delta,$$

where we assume that all clients share the same parameters, i.e., $w_i = w$, to simplify the derivation (typical assumption in FL).

Thus, the final gradient difference bound under federated adversarial scenarios is:

$$|\nabla F_S(w^{(1)}) - \nabla F_S(w^{(2)})\| \le L_w\|w^{(1)} - w^{(2)}\| + 2L_z\delta,$$

the proof of Lemma 1.1 is complete.

## C.2. Proof of Theorem 3.3

Inspired by (Hardt et al., 2016), denote by $S = \{z_{i,j}\}_{i\in[N], j\in[n]}$ and $S' = \{z'_{i,j}\}_{i\in[N], j\in[n]}$ two independent federated samples, where $z_{i,j}\sim\mathcal{D}_i$ and $z'_{i,j}\sim\mathcal{D}_i$ for each client $i$. For any $(i, j)$, let $S^{(j_i)}$ be identical to $S$ except that $z_{i,j}$ is replaced by $z'_{i,j}$, We deduce that

$$\mathbb{E}_{S,A}[F_S(A(S))] = \mathbb{E}_{S,A}\left[\frac{1}{Nn}\sum_{i=1}^{N}\sum_{j=1}^{n}f_i(A(S); z_{i,j})\right] \tag{11}$$

$$= \mathbb{E}_{S,S',A}\left[\frac{1}{Nn}\sum_{i=1}^{N}\sum_{j=1}^{n}f_i\Big(A(S^{(j_i)}); z'_{i,j}\Big)\right], \tag{12}$$

and

$$\mathbb{E}_{S,A}[F(A(S))] = \mathbb{E}_{S,A}\left[\frac{1}{N}\sum_{i=1}^{N}\mathbb{E}_{z\sim D_i}f_i(A(S); z)\right]$$

$$= \mathbb{E}_{S,S',A}\left[\frac{1}{Nn}\sum_{i=1}^{N}\sum_{j=1}^{n}f_i\big(A(S); z'_{i,j}\big)\right] \tag{13}$$

Subtracting (12) from (13) yields

$$\mathbb{E}_{S,A}[F(A(S)) - F_S(A(S))] = \mathbb{E}_{S,S',A}\left[\frac{1}{Nn}\sum_{i=1}^{N}\sum_{j=1}^{n}\Big(f_i(A(S); z'_{i,j}) - f_i(A(S^{(j_i)}); z'_{i,j})\Big)\right]. \tag{14}$$

Thus,

$$\left|\mathbb{E}_{S,A}[F(A(S)) - F_S(A(S))]\right| \leq \frac{1}{Nn} \sum_{i=1}^{N} \sum_{j=1}^{n} \left|\mathbb{E}_{S,S',A}\left[f_i(A(S); z'_{i,j}) - f_i(A(S^{(j_i)}); z'_{i,j})\right]\right| \tag{15}$$

$$\leq \frac{1}{Nn} \sum_{i=1}^{N} \sum_{j=1}^{n} \epsilon \tag{16}$$

$$= \epsilon, \tag{17}$$

where the second inequality follows from $\epsilon$-uniform stability, since $S$ and $S^{(j_i)}$ differ in exactly one example.

### C.3. Proof of Theorem 4.1 and 4.3

Let $S^{(j_i)} = S^{(n_N)} = \{S_i\}_{i=1}^{N-1} \cup S_N^{(n)}$. Define $\alpha_{i,j}^t = \left|\{m : z_{i,m}^t = z_{i,j}^t\}\right|, \quad \forall t \in \mathbb{N}, i \in \mathcal{M}_t, j \in [n], m \in [b_1]$. That is $\alpha_{i,j}^t$ is the number of samples equal to $z_{i,j}$ in the $t$-th global iteration for the $i$-th edge device. We have:

$$\mathbb{E}\left[\alpha_{i,j}^t\right] = \frac{b_1}{n}, \quad \mathbb{E}\left[(\alpha_{i,j}^t)^2\right] = \left(\mathbb{E}\left[\alpha_{i,j}^t\right]\right)^2 + \text{Var}\left(\alpha_{i,j}^t\right) = \frac{b_1}{n}\left(1 + \frac{b_1 - 1}{n}\right).$$

The update rule can be reformulated as

$$w^{t+1} = w^t - \frac{\eta_t}{b_1 M} \sum_{i \in \mathcal{M}_t} \sum_{j \in [n]} \alpha_{i,j}^t \nabla f_i(w_i^t; z_{i,m}^t). \tag{a}$$

From this formulation we obtain

$$\left\|w^{t+1} - \bar{w}^{t+1}\right\| = \left\|w^t - \bar{w}^t - \frac{\eta_t}{b_1 M} \sum_{i \in \mathcal{M}_t} \sum_{j \in [n]} \alpha_{i,j}^t \left(\nabla f_i(w_i^t; z_{i,j}^t) - \nabla f_i(\bar{w}_i^t; \bar{z}_{i,j}^t)\right)\right\|. \tag{b}$$

When $N \notin \mathcal{M}_t$, using Lemma 1.1 and $w_i^t = w^t$, we derive:

$$\left\|w^t - \bar{w}^t - \frac{\eta_t}{b_1 M} \sum_{i \in \mathcal{M}_t} \sum_{j \in [n]} \alpha_{i,j}^t \left(\nabla f_i(w_i^t; z_{i,j}^t) - \nabla f_i(\bar{w}_i^t; \bar{z}_{i,j}^t)\right)\right\|$$

$$\leq \left\|w^t - \bar{w}^t\right\| + \frac{\eta_t}{b_1 M} \sum_{i \in \mathcal{M}_t} \sum_{j \in [n]} \alpha_{i,j}^t \left\|\nabla f_i(w_i^t; z_{i,j}^t) - \nabla f_i(\bar{w}_i^t; z_{i,j}^t)\right\|$$

$$\leq \left\|w^t - \bar{w}^t\right\| + \frac{L_w \eta_t}{b_1 M} \sum_{i \in \mathcal{M}_t} \sum_{j \in [n]} \alpha_{i,j}^t \left\|w_i^t - \bar{w}_i^t\right\| + \frac{2 L_z \delta \eta_t}{b_1 M} \sum_{i \in \mathcal{M}_t} \sum_{j \in [n]} \alpha_{i,j}^t$$

$$\leq \left\|w^t - \bar{w}^t\right\| + \frac{L_w \eta_t}{b_1 M} \sum_{i \in \mathcal{M}_t} \sum_{j \in [n]} \alpha_{i,j}^t \left\|w^t - \bar{w}^t\right\| + \frac{2 L_z \delta \eta_t}{b_1 M} \sum_{i \in \mathcal{M}_t} \sum_{j \in [n]} \alpha_{i,j}^t$$

$$= \left(1 + \frac{L_w \eta_t}{b_1 M} \sum_{i \in \mathcal{M}_t} \sum_{j \in [n]} \alpha_{i,j}^t\right) \left\|w^t - \bar{w}^t\right\| + \frac{2 L_z \delta \eta_t}{b_1 M} \sum_{i \in \mathcal{M}_t} \sum_{j \in [n]} \alpha_{i,j}^t.$$

When $N \in \mathcal{M}_t$, let $P_t = \{(i,j) | i \in \mathcal{M}_t/\{N\}, j \in [n] \text{ or } i = N, j \in [n-1]\}$, then

$$\left\|w^t - \bar{w}^t - \frac{\eta_t}{b_1 M} \sum_{i \in \mathcal{M}_t} \sum_{j \in [n]} \alpha_{i,j}^t \left(\nabla f_i(w_i^t; z_{i,j}^t) - \nabla f_i(\bar{w}_i^t; \bar{z}_{i,j}^t)\right)\right\|$$

$$\leq \left\|w^t - \bar{w}^t\right\| + \frac{\eta_t}{b_1 M} \sum_{P_t} \alpha_{i,j}^t \|\nabla f_i(w_i^t; z_{i,j}^t) - \nabla f_i(\bar{w}_i^t; z_{i,j}^t)\|$$

$$+ \frac{\eta_t}{b_1 M} \alpha_{N,n}^t \|\nabla f_N\left(w_N^t; z_{N,n}^t\right) - \nabla f_N\left(\bar{w}_N^t; \bar{z}_{N,n}^t\right)\|$$

$$\leq \left\|w^t - \bar{w}^t\right\| + \frac{L_w \eta_t}{b_1 M} \sum_{P_t} \alpha_{i,j}^t \|w^t - \bar{w}^t\| + \frac{2 L_z \delta \eta_t}{b_1 M} \sum_{P_t} \alpha_{i,j}^t + \frac{2 \eta_t L}{b_1 M} \alpha_{N,n}^t$$

$$= \left(1 + \frac{L_w \eta_t}{b_1 M} \sum_{P_t} \alpha_{i,j}^t\right) \|w^t - \bar{w}^t\| + \frac{2L_z \delta \eta_t}{b_1 M} \sum_{P_t} \alpha_{i,j}^t + \frac{2\eta_t L}{b_1 M} \alpha_{N,n}^t.$$

Then, combining the above two inequalities, we obtain that

$$\|w^{t+1} - \bar{w}^{t+1}\|$$
$$\leq \frac{N-M}{N} \left( \left(1 + \frac{L_w \eta_t}{b_1 M} \sum_{i \in \mathcal{M}_t} \sum_{j \in [n]} \alpha_{i,j}^t\right) \|w^t - \bar{w}^t\| + \left(\frac{2L_z \delta \eta_t}{b_1 M} \sum_{i \in \mathcal{M}_t} \sum_{j \in [n]} \alpha_{i,j}^t\right) \right)$$
$$+ \frac{M}{N} \left( \left(1 + \frac{L_w \eta_t}{b_1 M} \sum_{P_t} \alpha_{i,j}^t\right) \|w^t - \bar{w}^t\| + \left(\frac{2L_z \delta \eta_t}{b_1 M} \sum_{P_t} \alpha_{i,j}^t + \frac{2\eta_t L}{b_1 M} \alpha_{N,n}^t\right) \right).$$

Define $J_i^t = \{z_{i,1}^t, \ldots, z_{i,b_1}^t\}, t \in \mathbb{N}, i \in [N]$. Taking conditional expectation w.r.t. $J_i^t$, we derive

$$\mathbb{E}_{J_i^t}\left[\|w^{t+1} - \bar{w}^{t+1}\|\right]$$
$$\leq \frac{N-M}{N} \left( \left(1 + \frac{L_w \eta_t}{b_1 M} \sum_{i \in \mathcal{M}_t} \sum_{j \in [n]} \mathbb{E}_{J_i^t}[\alpha_{i,j}^t]\right) \|w^t - \bar{w}^t\| + \left(\frac{2L_z \delta \eta_t}{b_1 M} \sum_{i \in \mathcal{M}_t} \sum_{j \in [n]} \mathbb{E}_{J_i^t}[\alpha_{i,j}^t]\right) \right)$$
$$+ \frac{M}{N} \left( \left(1 + \frac{L_w \eta_t}{b_1 M} \sum_{P_t} \mathbb{E}_{J_i^t}[\alpha_{i,j}^t]\right) \|w^t - \bar{w}^t\| + \left(\frac{2L_z \delta \eta_t}{b_1 M} \sum_{P_t} \mathbb{E}_{J_i^t}[\alpha_{i,j}^t] + \frac{2\eta_t L}{b_1 M} E_{J_N^t}[\alpha_{N,n}^t]\right) \right)$$
$$= \frac{N-M}{N}(1 + \eta_t L_w)\|w^t - \bar{w}^t\| + \frac{N-M}{N}(2L_z \delta \eta_t) + \frac{M}{N}(1 + \eta_t L_w)\|w^t - \bar{w}^t\| + \frac{M}{N}(2L_z \delta \eta_t) + \frac{2\eta_t L}{nN}$$
$$= (1 + \eta_t L_w)\|w^t - \bar{w}^t\| + 2L_z \delta \eta_t + \frac{2\eta_t L}{nN}.$$

Further taking expectation w.r.t. all randomness and using Lemmas 1.3, 1.4, we obtain that

$$\mathbb{E}[\|w^{t+1} - \bar{w}^{t+1}\|]$$
$$\leq (1 + \eta_t L_w)\|w^t - \bar{w}^t\| + 2L_z \delta \eta_t + \frac{2\eta_t L}{nN}.$$

Taking summation from $t = 1$ to $T - 1$, we deduce that

$$\mathbb{E}[\|w^T - \bar{w}^T\|]$$
$$\leq \sum_{t=1}^{T-1} \left( \prod_{s=t+1}^{T-1} (1 + \eta_s L_w) \right) \left( 2L_z \delta \eta_t + \frac{2\eta_t L}{nN} \right)$$
$$\leq \sum_{t=1}^{T-1} \exp\left( \sum_{s=t+1}^{T-1} \eta_s L_w \right) \left( 2L_z \delta \eta_t + \frac{2\eta_t L}{nN} \right)$$
$$\leq \sum_{t=1}^{T-1} \exp\left( \eta_1 L_w \sum_{s=1}^{T-1} s^{-1} \right) \left( 2L_z \delta \eta_t + \frac{2\eta_t L}{nN} \right)$$
$$= \exp\left( \eta_1 L_w \sum_{s=1}^{T-1} s^{-1} \right) \left( \frac{2L}{nN} + 2L_z \delta \right) \eta_1 \sum_{t=1}^{T-1} t^{-1}$$
$$\leq (e(T-1))^{\eta_1 L_w} \left( \frac{2L}{nN} + 2L_z \delta \right) \eta_1 \log(e(T-1))$$
$$\leq \mathcal{O}\left( \left((nN)^{-1}L + L_z \delta\right) T^{\frac{1}{2}} \log T \right),$$

where the second inequality is derived by $1 + x \leq e^x$ and the fourth inequality follows by Lemma 1.2. We obtain that

$$\left| \mathbb{E}\left[ F(w^T) - F_S(w^T) \right] \right|$$

$$\leq \frac{L}{nN} \sum_{i=1}^{N} \sum_{j=1}^{n} \mathbb{E}[\|w^T - \bar{w}^T\|] = L\mathbb{E}[\|w^T - \bar{w}^T\|]$$

$$\leq \mathcal{O}\left( L\left( (nN)^{-1}L + L_z\delta \right) T^{\frac{1}{2}} \log T \right).$$

The proof is complete.

### C.4. Proof of Theorem 4.6 and 4.8.

In federated adversarial learning with Non-convex losses, for any two neighboring federated datasets $S = \{S_1, \ldots, S_N\}$ and $S' = \{S_1, \ldots, S_i', \ldots, S_N\}$ differing in at most one sample (on client $i$). Let $S_i$ and $S_i'$ differ only in the $k$-th sample, and $\forall i \in [N]$, the Moreau envelope minimizers satisfy:

$$\left\| w_i(u; S_i) - w_i(u; S_i') \right\| \leq \frac{2L}{n(p - \ell)}$$

$$\left\| w(u; S) - w(u; S') \right\| = \frac{1}{N} \left\| w_i(u_i; S_i) - w_i(u_i; S_i') \right\| \leq \frac{2L}{Nn(p - l)}. \tag{a}$$

**Proof.** By the $(p - l)$-strongly convexity of $K_i(w_i, u_i; S_i)$, we have

$$(p - \ell)\left\| w_i(u_i; S_i) - w_i(u_i; S_i') \right\| \leq \left\| \nabla K_i(w_i(u_i; S_i), u_i; S_i) - \nabla K_i(w_i(u_i; S_i'), u_i; S_i) \right\|$$

$$\leq \left\| \nabla K_i(w_i(u_i; S_i), u_i; S_i) - \nabla K_i(w_i(u_i; S_i'), u_i; S_i') \right\|$$

$$+ \frac{1}{n}\left\| \nabla f_i(w_i(u_i; S_i'); z_{ik}) \right\| + \frac{1}{n}\left\| \nabla f_i(w_i(u_i; S_i'); z_{ik}') \right\|$$

$$= \frac{1}{n}\left\| \nabla f_i(w_i(u_i; S_i'); z_{ik}) - \nabla f_i(w_i(u_i; S_i'); z_{ik}') \right\|$$

$$\leq \frac{2L}{n}.$$

where the second inequality is due to the definition of $K_i(w_i, u_i; S_i)$, the third one is due to the first-order optimally condition,Since $w_i(u; S_i)$ and $w_i(u_i; S_i')$ are the minimizers of $K_i$ over $S_i$ and $S_i'$ respectively, their first-order optimality conditions are $\nabla K_i(w_i(u; S_i), u_i; S_i) = 0, \nabla K_i(w_i(u; S_i'), u; S_i') = 0$. And the last inequality is because of the bounded gradient of $f_i(w_i; z_i)$.

Based on the above conclusions, our proof proceeds in two steps. In Step 1, we establish a stability recursion. In Step 2, we unroll the resulting recursion using the uniform stability argument for non-smooth objectives.

**Step 1: build the recursion.**  Start from the one-step difference

$$\mathbb{E}\|u_S^{t+1} - u_{S'}^{t+1}\| = \mathbb{E}\left\| u_S^t - u_{S'}^t - \alpha_t\left( \nabla_u K(w_{N,S}^t, u_S^t; S) - \nabla_u K(w_{N,S'}^t, u_{S'}^t; S') \right) \right\|.$$

Add and Subtract Exact Gradient Terms

$$\mathbb{E}\left\| u_S^t - \alpha_t \nabla_u K(w_{N,S}^t, u_S^t; S) - u_{S'}^t + \nabla_u K(w_{N,S'}^t, u_{S'}^t; S') \right\|$$

$$\leq \mathbb{E}\| u_S^t - \alpha_t \nabla_u M(u_S^t; S) - u_{S'}^t + \alpha_t \nabla_u M(u_{S'}^t; S)\| + 2\alpha_t \mathbb{E}\|\nabla_u K(w_{N,S}^t, u_S^t; S) - \nabla_u M(u_S^t; S)\|$$

$$\leq \| u_S^t - u_{S'}^t - \alpha_t\left( \nabla_u M(u_S^t; S) - \nabla_u M(u_{S'}^t; S) \right)\| + 2\alpha_t p\varepsilon(\mathcal{A})$$

$$\leq \mathbb{E}\left\| u_S^t - u_{S'}^t - \alpha_t\left( \nabla_u M(u_S^t; S) + \nabla_u M(u_{S'}^t; S) \right)\right\| + \alpha_t\left\| \nabla_u M(u_{S'}^t; S) - \nabla_u M(u_{S'}^t; S') \right\| + 2\alpha_t p\varepsilon(\mathcal{A})$$

$$\leq \left\| u_S^t - u_{S'}^t \right\| + \alpha_t\left\| \nabla M(u_S^t; S) - \nabla M(u_{S'}^t; S) \right\| + \alpha_t\left\| \nabla M(u_{S'}^t; S') - \nabla M(u_{S'}^t; S) \right\| + 2\alpha_t p\,\varepsilon(\mathcal{A})$$

$$\leq (1 + \alpha_t L_w)\left\| u_S^t - u_{S'}^t \right\| + \alpha_t\left\| \nabla M(u_{S'}^t; S') - \nabla M(u_{S'}^t; S) \right\| + 2\alpha_t p\,\varepsilon(\mathcal{A}), \tag{b}$$

where the first step follows from the triangle inequality, the second step applies the assumed approximation guarantee of the inner solver for the inner minimization, the third and fourth steps again use the triangle inequality to separate the terms, and the final step invokes the $L_w$-Lipschitz continuity. $\varepsilon(\mathcal{A})$ is the optimization error of the inner problem.

Then,

$$\begin{aligned}
\alpha_t &\left\| \nabla M\left(u_{S'}^t; S'\right) - \nabla M\left(u_{S'}^t; S\right) \right\| \\
&= \alpha_t p \left\| u_{S'}^t - u_{S'}^t - w\left(u_{S'}^t, S\right) + w\left(u_{S'}^t, S'\right) \right\| \\
&\leq \frac{2Lp\,\alpha_t}{Nn\,(p-\ell)},
\end{aligned} \tag{c}$$

where the first inequality follows from the explicit representation of $\nabla M(u; S)$, and the final step applies (a).

Combining Eq. (b) and (c), we obtain the recursion

$$\begin{aligned}
\mathbb{E}\|u_S^{t+1} &- u_{S'}^{t+1}\| \\
&\leq (1 + \alpha_t L_w) \left\| u_S^t - u_{S'}^t \right\| + \frac{2Lp\alpha_t}{Nn(p-l)} + 2\alpha_t p\,\varepsilon(\mathcal{A}).
\end{aligned} \tag{d}$$

**Step 2: unwind the recursion.** Let

$$\Delta_t := \mathbb{E}\|u_S^t - u_{S'}^t\|, \qquad C := \frac{2Lp}{Nn(p-\ell)} + 2p\,\varepsilon(\mathcal{A}).$$

From (d), we have

$$\Delta_{t+1} \leq (1 + \alpha_t L_w)\Delta_t + C\alpha_t.$$

Taking summation from $t = 1$ to $T - 1$, we deduce that

$$\begin{aligned}
\Delta_T &\leq \sum_{t=1}^{T-1} \left( \prod_{s=t+1}^{T-1} (1 + \alpha_s L_w) \right) C\alpha_t \\
&\leq \sum_{t=1}^{T-1} \exp\left( \sum_{s=t+1}^{T-1} \alpha_s L_w \right) C\alpha_t \\
&\leq \sum_{t=1}^{T-1} \exp\left( \alpha_1 L_w \sum_{s=1}^{T-1} s^{-1} \right) C\alpha_t \qquad \text{(take } \alpha_t = \alpha_1 t^{-1}) \\
&= \exp\left( \alpha_1 L_w \sum_{s=1}^{T-1} s^{-1} \right) C\alpha_1 \sum_{t=1}^{T-1} t^{-1} \\
&\leq (e(T-1))^{\alpha_1 L_w} C\alpha_1 \log(e(T-1)) \\
&\leq \mathcal{O}\left( C\,T^{\alpha_1 L_w} \log T \right).
\end{aligned}$$

In particular, if $\alpha_1 = \frac{1}{2L_w}$, then

$$\Delta_T \leq \mathcal{O}\left( C\,T^{\frac{1}{2}} \log T \right).$$

Assume the inner algorithm $\mathcal{A}$ satisfies

$$\varepsilon(\mathcal{A}) = \mathcal{O}\left( \frac{1}{(p-\ell)n} \right).$$

Recall

$$C := \frac{2Lp}{Nn(p-\ell)} + 2p\,\varepsilon(\mathcal{A}) = \mathcal{O}\left( \frac{Lp}{Nn(p-\ell)} + \frac{p}{(p-\ell)n} \right) = \mathcal{O}\left( \frac{p}{(p-\ell)} \left( \frac{L}{Nn} + \frac{1}{n} \right) \right).$$

With $\alpha_t = \alpha_1 t^{-1}$ and $\alpha_1 = \frac{1}{2L_w}$, we have

$$\Delta_T = \mathbb{E}\|u_S^T - u_{S'}^T\| \leq \mathcal{O}\left( C\,T^{\frac{1}{2}} \log T \right) = \mathcal{O}\left( \frac{p}{(p-\ell)} \left( \frac{L}{Nn} + \frac{1}{n} \right) T^{\frac{1}{2}} \log T \right).$$

Therefore,

$$\mathcal{E}_{\text{gen}} \leq L\,\Delta_T \leq \mathcal{O}\left( \frac{Lp}{(p-\ell)} \left( \frac{L}{Nn} + \frac{1}{n} \right) T^{\frac{1}{2}} \log T \right).$$

The proof is complete.

## C.5. Proof of Theorem 4.10 and 4.11

Let $S^{(j_i)} = S^{(n_N)} = \{S_i\}_{i=1}^{N-1} \cup S_N^{(n)}$. Define $\alpha_{i,j}^t = \left|\{m : z_{i,m}^t = z_{i,j}^t\}\right|, \forall t \in \mathbb{N}, i \in \mathcal{M}_t, j \in [n], m \in [b_1]$. That is $\alpha_{i,j}^t$ is the number of samples that are equal to $z_{i,j}$ in the $t-th$ global iteration for the $i-the$dge device. It is obvious that $\mathbb{E}\left[\alpha_{ij}^t\right] = b_1/n, \mathbb{E}\left[(\alpha_{i,j}^t)^2\right] = \left(\mathbb{E}\left[\alpha_{i,j}^t\right]\right)^2 + Var\left(\alpha_{i,j}^t\right) = \frac{b_1}{n}\left(1 + \frac{b_1-1}{n}\right)$. Then, the update can be reformulated as

$$w^{t+1} = w^t - \frac{\eta_t}{b_1 M} \sum_{i \in \mathcal{M}_t} \sum_{j \in [n]}^{b_1} \alpha_{i,j}^t \tilde{\nabla} f_i\left(w_i^t; z_{i,j}^t, \{v_{il}^t\}_{l=1}^{b_2}, \mu\right).$$

For the sake of simplicity, we denote $\tilde{\nabla} f_i\left(w_i^t; z_{i,j}^t, \{v_{il}^t\}_{l=1}^{b_2}, \mu\right)$ as $\tilde{\nabla} f_i(w_i^t; z_{i,j}^t)$. According to the new formulation, we can get

$$\left\|w^{t+1} - \bar{w}^{t+1}\right\|$$

$$= \left\|w^t - \bar{w}^t - \frac{\eta_t}{b_1 M} \sum_{i \in \mathcal{M}_t} \sum_{j \in [n]} \alpha_{i,j}^t \left(\tilde{\nabla} f_i\left(w_i^t; z_{i,j}^t\right) - \tilde{\nabla} f_i\left(\bar{w}_i^t; \bar{z}_{i,j}^t\right)\right)\right\|$$

$$\leq \left\|w^t - \bar{w}^t - \frac{\eta_t}{b_1 M} \sum_{i \in \mathcal{M}_t} \sum_{j \in [n]} \alpha_{i,j}^t \left(\nabla f_i\left(w_i^t; z_{i,j}^t\right) - \nabla f_i\left(\bar{w}_i^t; \bar{z}_{i,j}^t\right)\right)\right\| \tag{16}$$

$$+ \left\|\frac{\eta_t}{b_1 M} \sum_{i \in \mathcal{M}_t} \sum_{j \in [n]} \alpha_{i,j}^t \left(\tilde{\nabla} f_i\left(w_i^t; z_{i,j}^t\right) - \tilde{\nabla} f_i\left(\bar{w}_i^t; \bar{z}_{i,j}^t\right) - \nabla f_i\left(w_i^t; z_{i,j}^t\right) + \nabla f_i(\bar{w}_i^t; \bar{z}_{i,j}^t)\right)\right\|. \tag{17}$$

Considering the possibility of choosing a client who has a disturbed sample, we carry out the following discussion. When $N \notin \mathcal{M}_t$, we using Lemma 1.1, the fact that $w_i^t = w^t$ to get

$$\left\|w^t - \bar{w}^t - \frac{\eta_t}{b_1 M} \sum_{i \in \mathcal{M}_t} \sum_{j \in [n]} \alpha_{i,j}^t \left(\nabla f_i\left(w_i^t; z_{i,j}^t\right) - \nabla f_i\left(\bar{w}_i^t; \bar{z}_{i,j}^t\right)\right)\right\|$$

$$\leq \left\|w^t - \bar{w}^t\right\| + \frac{\eta_t}{b_1 M} \sum_{i \in \mathcal{M}_t} \sum_{j \in [n]} \alpha_{i,j}^t \left\|\nabla f_i(w_i^t; z_{i,j}^t) - \nabla f_i(\bar{w}_i^t; z_{i,j}^t)\right\|$$

$$\leq \left\|w^t - \bar{w}^t\right\| + \frac{L_w \eta_t}{b_1 M} \sum_{i \in \mathcal{M}_t} \sum_{j \in [n]} \alpha_{i,j}^t \left\|w_i^t - \bar{w}_i^t\right\| + \frac{2L_z \delta \eta_t}{b_1 M} \sum_{i \in \mathcal{M}_t} \sum_{j \in [n]} \alpha_{i,j}^t$$

$$\leq \left\|w^t - \bar{w}^t\right\| + \frac{L_w \eta_t}{b_1 M} \sum_{i \in \mathcal{M}_t} \sum_{j \in [n]} \alpha_{i,j}^t \left\|w^t - \bar{w}^t\right\| + \frac{2L_z \delta \eta_t}{b_1 M} \sum_{i \in \mathcal{M}_t} \sum_{j \in [n]} \alpha_{i,j}^t$$

$$= \left(1 + \frac{L_w \eta_t}{b_1 M} \sum_{i \in \mathcal{M}_t} \sum_{j \in [n]} \alpha_{i,j}^t\right) \left\|w^t - \bar{w}^t\right\| + \frac{2L_z \delta \eta_t}{b_1 M} \sum_{i \in \mathcal{M}_t} \sum_{j \in [n]} \alpha_{i,j}^t.$$

and

$$\left\|\frac{\eta_t}{b_1 M} \sum_{i \in \mathcal{M}_t} \sum_{j \in [n]} \alpha_{i,j}^t \left(\tilde{\nabla} f_i\left(w_i^t; z_{i,j}^t\right) - \tilde{\nabla} f_i\left(\bar{w}_i^t; \bar{z}_{i,j}^t\right) - \nabla f_i\left(w_i^t; z_{i,j}^t\right) + \nabla f_i(\bar{w}_i^t; \bar{z}_{i,j}^t)\right)\right\|$$

$$\leq \frac{\eta_t}{b_1 M} \sum_{i \in \mathcal{M}_t} \sum_{j \in [n]} \alpha_{i,j}^t \left\|\tilde{\nabla} f_i\left(w_i^t; z_{i,j}^t\right) - \tilde{\nabla} f_i\left(\bar{w}_i^t; z_{i,j}^t\right) - \nabla f_i\left(w_i^t; z_{i,j}^t\right) + \nabla f_i(\bar{w}_i^t; z_{i,j}^t)\right\|$$

$$= \frac{\eta_t}{b_1 M} \sum_{i \in \mathcal{M}_t} \sum_{j \in [n]} \alpha_{i,j}^t \left\|\frac{1}{b_2} \sum_{l=1}^{b_2} \left(\langle \nabla f_i\left(w_i^t; z_{i,j}^t\right) - \nabla f_i\left(\bar{w}_i^t; z_{i,j}^t\right), v_{il}^t\rangle v_{il}^t\right.\right.$$

$$\left.\left. + \left(\frac{\mu}{2}\left(v_{il}^t\right)^\top \nabla_{w_i}^2 f_i\left(w_i; z_{i,j}^t\right)\Big|_{w_i = w_{il}^{t*}} v_{il}^t\right) v_{il}^t - \left(\frac{\mu}{2}\left(v_{il}^t\right)^\top \nabla_{w_i}^2 f_i\left(w_i; z_{i,j}^t\right)\Big|_{w_i = w_{il}^{t\dagger}} v_{il}^t\right) v_{il}^t\right)\right\|$$

$$
\left. - \nabla f_i\left(w_i^t; z_{i,j}^t\right) + \nabla f_i\left(\bar{w}_i^t; z_{i,j}^t\right) \right\|
$$

$$
= \frac{\eta_t}{b_1 M} \sum_{i \in \mathcal{M}_t} \sum_{j \in [n]} \alpha_{i,j}^t \Bigg( \left\| \frac{1}{b_2} \sum_{l=1}^{b_2} \left( \left( \frac{\mu}{2} \left(v_{il}^t\right)^\top \nabla_{w_i}^2 f_i\left(w_i; z_{i,j}^t\right) \Big|_{w_i = w_{il}^{t*}} v_{il}^t \right) v_{il}^t \right. \right.
$$

$$
\left. - \left( \frac{\mu}{2} \left(v_{il}^t\right)^\top \nabla_{w_i}^2 f_i\left(w_i; z_{i,j}^t\right) \Big|_{w_i = w_{il}^{t\dagger}} v_{il}^t \right) v_{il}^t \right) \right\|
$$

$$
+ \left\| \frac{1}{b_2} \sum_{l=1}^{b_2} \left\langle \nabla f_i\left(w_i^t; z_{i,j}^t\right) - \nabla f_i\left(\bar{w}_i^t; z_{i,j}^t\right), v_{il}^t \right\rangle v_{il}^t - \nabla f_i\left(w_i^t; z_{i,j}^t\right) + \nabla f_i\left(\bar{w}_i^t; z_{i,j}^t\right) \right\| \Bigg)
$$

$$
\leq \frac{\eta_t}{b_1 M} \sum_{i \in \mathcal{M}_t} \sum_{j \in [n]} \alpha_{i,j}^t \Bigg( \frac{2}{b_2} \sum_{l=1}^{b_2} \frac{\mu L_w}{2} \left\| v_{il}^t \right\|^3
$$

$$
+ \left\| \frac{1}{b_2} \sum_{l=1}^{b_2} \left\langle \nabla f_i\left(w_i^t; z_{i,j}^t\right) - \nabla f_i\left(\bar{w}_i^t; z_{i,j}^t\right), v_{il}^t \right\rangle v_{il}^t - \nabla f_i\left(w_i^t; z_{i,j}^t\right) + \nabla f_i\left(\bar{w}_i^t; z_{i,j}^t\right) \right\| \Bigg)
$$

$$
\leq \frac{\eta_t}{b_1 M} \sum_{i \in \mathcal{M}_t} \sum_{j \in [n]} \alpha_{i,j}^t \Bigg( \frac{\mu L_w}{b_2} \sum_{l=1}^{b_2} \left\| v_{il}^t \right\|^3
$$

$$
+ \left\| \frac{1}{b_2} \sum_{l=1}^{b_2} \left\langle \nabla f_i\left(w_i^t; z_{i,j}^t\right) - \nabla f_i\left(\bar{w}_i^t; z_{i,j}^t\right), v_{il}^t \right\rangle v_{il}^t - \nabla f_i\left(w_i^t; z_{i,j}^t\right) + \nabla f_i\left(\bar{w}_i^t; z_{i,j}^t\right) \right\| \Bigg).
$$

When $N \in \mathcal{M}_t$, let $P_t = \{(i,j) | i \in \mathcal{M}_t / \{N\}, j \in [n] \text{ or } i = N, j \in [n-1]\}$, then

$$
\left\| w^t - \bar{w}^t - \frac{\eta_t}{b_1 M} \sum_{i \in \mathcal{M}_t} \sum_{j \in [n]} \alpha_{i,j}^t \left( \nabla f_i\left(w_i^t; z_{i,j}^t\right) - \nabla f_i\left(\bar{w}_i^t; \bar{z}_{i,j}^t\right) \right) \right\|
$$

$$
\leq \left\| w^t - \bar{w}^t \right\| + \frac{\eta_t}{b_1 M} \sum_{P_t} \alpha_{i,j}^t \left\| \left( \nabla f_i(w_i^t; z_{i,j}^t) - \nabla f_i(\bar{w}_i^t; z_{i,j}^t) \right) \right\|
$$

$$
+ \frac{\eta_t}{b_1 M} \alpha_{N,n}^t \left\| \nabla f_N\left(w_N^t; z_{N,n}^t\right) - \nabla f_N\left(\bar{w}_N^t; \bar{z}_{N,n}^t\right) \right\|
$$

$$
\leq \left\| w^t - \bar{w}^t \right\| + \frac{L_w \eta_t}{b_1 M} \sum_{P_t} \alpha_{i,j}^t \left\| w^t - \bar{w}^t \right\| + \frac{2 L_z \delta \eta_t}{b_1 M} \sum_{P_t} \alpha_{i,j}^t + \frac{2 \eta_t L}{b_1 M} \alpha_{N,n}^t
$$

$$
= \left( 1 + \frac{L_w \eta_t}{b_1 M} \sum_{P_t} \alpha_{i,j}^t \right) \left\| w^t - \bar{w}^t \right\| + \frac{2 L_z \delta \eta_t}{b_1 M} \sum_{P_t} \alpha_{i,j}^t + \frac{2 \eta_t L}{b_1 M} \alpha_{N,n}^t.
$$

and

$$
\left\| \frac{\eta_t}{b_1 M} \sum_{i \in \mathcal{M}_t} \sum_{j \in [n]} \alpha_{i,j}^t \left( \tilde{\nabla} f_i\left(w_i^t; z_{i,j}^t\right) - \tilde{\nabla} f_i\left(\bar{w}_i^t; \bar{z}_{i,j}^t\right) - \nabla f_i\left(w_i^t; z_{i,j}^t\right) + \nabla f_i(\bar{w}_i^t; \bar{z}_{i,j}^t) \right) \right\|
$$

$$
\leq \frac{\eta_t}{b_1 M} \sum_{P_t} \alpha_{i,j}^t \left\| \tilde{\nabla} f_i\left(w_i^t; z_{i,j}^t\right) - \tilde{\nabla} f_i\left(\bar{w}_i^t; z_{i,j}^t\right) - \nabla f_i\left(w_i^t; z_{i,j}^t\right) + \nabla f_i\left(\bar{w}_i^t; z_{i,j}^t\right) \right\|
$$

$$
+ \frac{\eta_t}{b_1 M} \alpha_{N,n}^t \left\| \tilde{\nabla} f_N\left(w_N^t; z_{N,n}^t\right) - \tilde{\nabla} f_N\left(\bar{w}_N^t; \bar{z}_{N,n}^t\right) - \nabla f_N\left(w_N^t; z_{N,n}^t\right) + \nabla f_N\left(\bar{w}_N^t; \bar{z}_{N,n}^t\right) \right\|
$$

$$
\leq \frac{\eta_t}{b_1 M} \sum_{i \in \mathcal{M}_t} \sum_{j \in [n]} \alpha_{i,j}^t \frac{\mu L_w}{b_2} \sum_{l=1}^{b_2} \left\| v_{il}^t \right\|^3
$$

$$+ \frac{\eta_t}{b_1 M} \sum_{P_t} \alpha_{i,j}^t \left\| \frac{1}{b_2} \sum_{l=1}^{b_2} \langle \nabla f_i\left(w_i^t; z_{i,j}^t\right) - \nabla f_i\left(\bar{w}_i^t; z_{i,j}^t\right), v_{il}^t \rangle v_{il}^t - \nabla f_i\left(w_i^t; z_{i,j}^t\right) + \nabla f_i\left(\bar{w}_i^t; z_{i,j}^t\right) \right\|$$

$$+ \frac{\eta_t}{b_1 M} \alpha_{N,n}^t \left\| \frac{1}{b_2} \sum_{l=1}^{b_2} \langle \nabla f_N\left(w_N^t; z_{N,n}^t\right) - \nabla f_N\left(\bar{w}_N^t; \bar{z}_{N,n}^t\right), v_{Nl}^t \rangle v_{Nl}^t - \nabla f_N\left(w_N^t; z_{N,n}^t\right) + \nabla f_N\left(\bar{w}_N^t; z_{N,n}^t\right) \right\|.$$

Then, combining the above four inequalities, we obtain that

$$\left\| w^{t+1} - \bar{w}^{t+1} \right\|$$

$$\leq \frac{N-M}{N} \Bigg( \left( 1 + \frac{L_w \eta_t}{b_1 M} \sum_{i \in \mathcal{M}_t} \sum_{j \in [n]} \alpha_{i,j}^t \right) \left\| w^t - \bar{w}^t \right\| + \frac{2 L_z \delta \eta_t}{b_1 M} \sum_{i \in \mathcal{M}_t} \sum_{j \in [n]} \alpha_{i,j}^t + \frac{\eta_t}{b_1 M} \sum_{i \in \mathcal{M}_t} \sum_{j \in [n]}$$

$$\alpha_{i,j}^t \left( \frac{\mu L_w}{b_2} \sum_{l=1}^{b_2} \|v_{il}^t\|^3 + \left\| \frac{1}{b_2} \sum_{l=1}^{b_2} \langle \nabla f_i\left(w_i^t; z_{i,j}^t\right) - \nabla f_i\left(\bar{w}_i^t; z_{i,j}^t\right), v_{il}^t \rangle v_{il}^t - \nabla f_i\left(w_i^t; z_{i,j}^t\right) + \nabla f_i\left(\bar{w}_i^t; z_{i,j}^t\right) \right\| \right) \Bigg)$$

$$+ \frac{M}{N} \Bigg( \left( 1 + \frac{L_w \eta_t}{b_1 M} \sum_{P_t} \alpha_{i,j}^t \right) \left\| w^t - \bar{w}^t \right\| + \frac{2 L_z \delta \eta_t}{b_1 M} \sum_{P_t} \alpha_{i,j}^t + \frac{2\eta_t L}{b_1 M} \alpha_{N,n}^t + \frac{\eta_t}{b_1 M} \sum_{i \in \mathcal{M}_t} \sum_{j \in [n]} \alpha_{i,j}^t \frac{\mu L_w}{b_2} \sum_{l=1}^{b_2} \|v_{il}^t\|^3$$

$$+ \frac{\eta_t}{b_1 M} \sum_{P_t} \alpha_{i,j}^t \left\| \frac{1}{b_2} \sum_{l=1}^{b_2} \langle \nabla f_i\left(w_i^t; z_{i,j}^t\right) - \nabla f_i\left(\bar{w}_i^t; z_{i,j}^t\right), v_{il}^t \rangle v_{il}^t - \nabla f_i\left(w_i^t; z_{i,j}^t\right) + \nabla f_i\left(\bar{w}_i^t; z_{i,j}^t\right) \right\|$$

$$+ \frac{\eta_t}{b_1 M} \alpha_{N,n}^t \left\| \frac{1}{b_2} \sum_{l=1}^{b_2} \langle \nabla f_N\left(w_N^t; z_{N,n}^t\right) - \nabla f_N\left(\bar{w}_N^t; \bar{z}_{N,n}^t\right), v_{Nl}^t \rangle v_{Nl}^t - \nabla f_N\left(w_N^t; z_{N,n}^t\right) + \nabla f_N\left(\bar{w}_N^t; \bar{z}_{N,n}^t\right) \right\| \Bigg).$$

Define $J_i^t = \{z_{i,1}^t, ..., z_{i,b_1}^t\}, t \in \mathbb{N}, i \in [N]$. Taking conditional expectation w.r.t. $J_i^t$, we derive

$$\mathbb{E}_{J_i^t}\left[ \left\| w^{t+1} - \bar{w}^{t+1} \right\| \right]$$

$$\leq \frac{N-M}{N} \Bigg( \left( 1 + \frac{L_w \eta_t}{b_1 M} \sum_{i \in \mathcal{M}_t} \sum_{j \in [n]} \mathbb{E}_{J_i^t}\left[\alpha_{i,j}^t\right] \right) \left\| w^t - \bar{w}^t \right\| + \frac{2 L_z \delta \eta_t}{b_1 M} \sum_{i \in \mathcal{M}_t} \sum_{j \in [n]} \mathbb{E}_{J_i^t}\left[\alpha_{i,j}^t\right] + \frac{\eta_t}{b_1 M} \sum_{i \in \mathcal{M}_t} \sum_{j \in [n]}$$

$$\mathbb{E}_{J_i^t}\left[\alpha_{i,j}^t\right] \left( \frac{\mu L_w}{b_2} \sum_{l=1}^{b_2} \|v_{il}^t\|^3 + \left\| \frac{1}{b_2} \sum_{l=1}^{b_2} \langle \nabla f_i\left(w_i^t; z_{i,j}^t\right) - \nabla f_i\left(\bar{w}_i^t; z_{i,j}^t\right), v_{il}^t \rangle v_{il}^t - \nabla f_i\left(w_i^t; z_{i,j}^t\right) \right. \right.$$

$$\left. \left. + \nabla f_i\left(\bar{w}_i^t; z_{i,j}^t\right) \right\| \right) \Bigg) + \frac{M}{N} \Bigg( \left( 1 + \frac{L_w \eta_t}{b_1 M} \sum_{P_t} \mathbb{E}_{J_i^t}\left[\alpha_{i,j}^t\right] \right) \left\| w^t - \bar{w}^t \right\| + \frac{2 L_z \delta \eta_t}{b_1 M} \mathbb{E}_{J_i^t}\left[\alpha_{i,j}^t\right]$$

$$+ \frac{2\eta_t L}{b_1 M} \mathbb{E}_{J_N^t}\left[\alpha_{N,n}^t\right] + \frac{\eta_t}{b_1 M} \sum_{i \in \mathcal{M}_t} \sum_{j \in [n]} \mathbb{E}_{J_i^t}\left[\alpha_{i,j}^t\right] \frac{\mu L_w}{b_2} \sum_{l=1}^{b_2} \|v_{il}^t\|^3$$

$$+ \frac{\eta_t}{b_1 M} \sum_{P_t} \mathbb{E}_{J_i^t}\left[\alpha_{i,j}^t\right] \left\| \frac{1}{b_2} \sum_{l=1}^{b_2} \langle \nabla f_i\left(w_i^t; z_{i,j}^t\right) - \nabla f_i\left(\bar{w}_i^t; z_{i,j}^t\right), v_{il}^t \rangle v_{il}^t - \nabla f_i\left(w_i^t; z_{i,j}^t\right) + \nabla f_i\left(\bar{w}_i^t; z_{i,j}^t\right) \right\|$$

$$+ \frac{\eta_t}{b_1 M} \mathbb{E}_{J_N^t}\left[\alpha_{N,n}^t\right] \left\| \frac{1}{b_2} \sum_{l=1}^{b_2} \langle \nabla f_N\left(w_N^t; z_{N,n}^t\right) - \nabla f_N\left(\bar{w}_N^t; \bar{z}_{N,n}^t\right), v_{Nl}^t \rangle v_{Nl}^t - \nabla f_N\left(w_N^t; z_{N,n}^t\right) + \nabla f_N\left(\bar{w}_N^t; z_{N,n}^t\right) \right\| \Bigg)$$

$$= \frac{N-M}{N}(1 + \eta_t L_w) \left\| w^t - \bar{w}^t \right\| + \frac{M}{N}(1 + \eta_t L_w) \left\| w^t - \bar{w}^t \right\| + 2 L_z \delta \eta_t + \frac{M}{N}(2 L_z \delta \eta_t) + \frac{2\eta_t L}{nN} + \frac{\mu \eta_t L_w}{b_2} \sum_{l=1}^{b_2} \|v_{il}^t\|^3$$

$$+ \frac{N-M}{N} \eta_t \left\| \frac{1}{b_2} \sum_{l=1}^{b_2} \langle \nabla f_i\left(w_i^t; z_{i,j}^t\right) - \nabla f_i\left(\bar{w}_i^t; z_{i,j}^t\right), v_{il}^t \rangle v_{il}^t - \nabla f_i\left(w_i^t; z_{i,j}^t\right) + \nabla f_i\left(\bar{w}_i^t; z_{i,j}^t\right) \right\|$$

$$+ \frac{M}{N}\eta_t \left\| \frac{1}{b_2}\sum_{l=1}^{b_2} \left\langle \nabla f_i\left(w_i^t; z_{i,j}^t\right) - \nabla f_i\left(\bar{w}_i^t; z_{i,j}^t\right), v_{il}^t \right\rangle v_{il}^t - \nabla f_i\left(w_i^t; z_{i,j}^t\right) + \nabla f_i\left(\bar{w}_i^t; z_{i,j}^t\right) \right\|$$

$$+ \frac{\eta_t}{nN} \left\| \frac{1}{b_2}\sum_{l=1}^{b_2} \left\langle \nabla f_N\left(w_N^t; z_{N,n}^t\right) - \nabla f_N\left(\bar{w}_N^t; \bar{z}_{N,n}^t\right), v_{Nl}^t \right\rangle v_{Nl}^t - \nabla f_N\left(w_N^t; z_{N,n}^t\right) + \nabla f_N\left(\bar{w}_N^t; z_{N,n}^t\right) \right\|$$

$$\leq (1 + \eta_t L_w)\left\| w^t - \bar{w}^t \right\| + (1 + \frac{M}{N})2L_z\delta\eta_t + \frac{2\eta_t L}{nN} + \frac{\mu\eta_t L_w}{b_2}\sum_{l=1}^{b_2}\left\| v_{il}^t \right\|^3$$

$$+ \eta_t \left\| \frac{1}{b_2}\sum_{l=1}^{b_2} \left\langle \nabla f_i\left(w_i^t; z_{i,j}^t\right) - \nabla f_i\left(\bar{w}_i^t; z_{i,j}^t\right), v_{il}^t \right\rangle v_{il}^t - \nabla f_i\left(w_i^t; z_{i,j}^t\right) + \nabla f_i\left(\bar{w}_i^t; z_{i,j}^t\right) \right\|$$

$$+ \frac{\eta_t}{nN} \left\| \frac{1}{b_2}\sum_{l=1}^{b_2} \left\langle \nabla f_N\left(w_N^t; z_{N,n}^t\right) - \nabla f_N\left(\bar{w}_N^t; \bar{z}_{N,n}^t\right), v_{Nl}^t \right\rangle v_{Nl}^t - \nabla f_N\left(w_N^t; z_{N,n}^t\right) + \nabla f_N\left(\bar{w}_N^t; z_{N,n}^t\right) \right\|.$$

Further taking expectation w.r.t. all randomness and using Lemmas 1.3, 1.4, we obtain that

$$\mathbb{E}\left[\left\| w^{t+1} - \bar{w}^{t+1} \right\|\right]$$

$$\leq (1 + \eta_t L_w)\mathbb{E}\left[\left\| w^t - \bar{w}^t \right\|\right] + (1 + \frac{M}{N})2L_z\delta\eta_t + \frac{2\eta_t L}{nN} + \mu\eta_t L_w\mathbb{E}\left[\left\| v_{il}^t \right\|^3\right]$$

$$+ \eta_t\mathbb{E}\left[\left\| \frac{1}{b_2}\sum_{l=1}^{b_2} \left\langle \nabla f_i\left(w_i^t; z_{i,j}^t\right) - \nabla f_i\left(\bar{w}_i^t; z_{i,j}^t\right), v_{il}^t \right\rangle v_{il}^t - \nabla f_i\left(w_i^t; z_{i,j}^t\right) + \nabla f_i\left(\bar{w}_i^t; z_{i,j}^t\right) \right\|\right]$$

$$+ \frac{\eta_t}{nN}\mathbb{E}\left[\left\| \frac{1}{b_2}\sum_{l=1}^{b_2} \left\langle \nabla f_N\left(w_N^t; z_{N,n}^t\right) - \nabla f_N\left(\bar{w}_N^t; \bar{z}_{N,n}^t\right), v_{Nl}^t \right\rangle v_{Nl}^t - \nabla f_N\left(w_N^t; z_{N,n}^t\right) + \nabla f_N\left(\bar{w}_N^t; z_{N,n}^t\right) \right\|\right]$$

$$\leq (1 + \eta_t L_w)\mathbb{E}\left[\left\| w^t - \bar{w}^t \right\|\right] + (1 + \frac{M}{N})2L_z\delta\eta_t + \frac{2\eta_t L}{nN} + \frac{d\mu\eta_t L_w}{d+3} + \eta_t\sqrt{\frac{d}{b_2}}\mathbb{E}\left[\left\| \nabla f_i\left(w_i^t; z_{i,j}^t\right) - \nabla f_i\left(\bar{w}_i^t; z_{i,j}^t\right) \right\|\right]$$

$$+ \frac{\eta_t}{nN}\sqrt{\frac{d}{b_2}}\mathbb{E}\left[\left\| \nabla f_N\left(w_N^t; z_{N,j}^t\right) - \nabla f_N\left(\bar{w}_N^t; z_{N,j}^t\right) \right\|\right]$$

$$\leq \left(1 + \left(1 + \sqrt{\frac{d}{b_2}}\right)\eta_t L_w\right)\mathbb{E}\left[\left\| w^t - \bar{w}^t \right\|\right] + \left(\frac{2L}{nN} + \mu L_w + 2L_z\delta + \frac{2ML_z\delta}{N} + 2L_z\delta\sqrt{\frac{d}{b_2}} + \frac{2L}{nN}\sqrt{\frac{d}{b_2}}\right)\eta_t.$$

Let $a_1 = \left(1 + \sqrt{\frac{d}{b_2}}\right)L_w$ and $a_2 = \left(\frac{2L}{nN} + \mu L_w + 2L_z\delta + \frac{2ML_z\delta}{N} + 2L_z\delta\sqrt{\frac{d}{b_2}} + \frac{2L}{nN}\sqrt{\frac{d}{b_2}}\right)$. Taking summation from $t = 1$ to $T - 1$, we deduce that

$$\mathbb{E}\left[\left\| w^T - \bar{w}^T \right\|\right]$$

$$\leq \sum_{t=1}^{T-1}\left(\prod_{s=t+1}^{T-1}(1 + a_1\eta_s)\right)a_2\eta_t$$

$$\leq \sum_{t=1}^{T-1}\exp\left(\sum_{s=t+1}^{T-1}a_1\eta_s\right)a_2\eta_t$$

$$\leq \sum_{t=1}^{T-1}\exp\left(a_1\eta_1\sum_{s=1}^{T-1}s^{-1}\right)a_2\eta_t$$

$$= \exp\left(a_1\eta_1\sum_{s=1}^{T-1}s^{-1}\right)a_2\eta_1\sum_{t=1}^{T-1}t^{-1}$$

$$\leq (e(T-1))^{a_1\eta_1}a_2\eta_1\log(e(T-1))$$

$$\leq \mathcal{O}\left(\left((nN)^{-1}L + L_z\delta + \mu\right)T^{\frac{1}{2}}\log T\right),$$

where the second inequality is derived by $1 + x \le e^x$ and the fourth inequality follows by Lemma 1.2. We obtain that

$$
\begin{aligned}
&\left| \mathbb{E}\left[ F(w^T) - F_S(w^T) \right] \right| \\
\le& \frac{L}{nN} \sum_{i=1}^{N} \sum_{j=1}^{n} \mathbb{E}[\|w^T - \bar{w}^T\|] = L\mathbb{E}[\|w^T - \bar{w}^T\|] \\
\le& \mathcal{O}\left( L\left( (nN)^{-1}L + \mu + L_z\delta \right) T^{\frac{1}{2}} \log T \right).
\end{aligned}
$$

The proof is complete.

## D. Additional Works

In this section, we give our FAL algorithms and additional experimental results.

Table3 : Summary of federated adversarial learning algorithms.

| Algorithm | Reference | Adversarial Training | Generalization Guarantee |
|---|---|---|---|
| FAT | (Zizzo et al., 2020) | PGD-based | No |
| FedDynAT | (Shah et al., 2021) | PGD-based + dynamic local steps | No |
| FedBVA | (Zhou et al., 2022) | PGD-based + Bias-Variance | No |
| FedRBN | (Hong et al., 2023) | PGD-based + BN propagation | Yes |
| DBFAT | (Zhang et al., 2023) | PGD-based + boundary regularization | No |
| SFAT | (Zhu et al., 2023) | PGD-based + slack mechanism | No |

### D.1. Additional Experiments

We adopt a convolutional neural network (CNN) with two convolutional layers (10 and 20 filters, kernel size 5), each followed by ReLU activation, max pooling, and dropout. Features are fed into a fully connected layer with 50 units and dropout, and a final output layer for classification. All models are trained with an initial learning rate of 0.01 (decayed during training), batch size 10, and local updates are performed for 5 epochs on each client per communication round. The MNIST dataset is partitioned among 100 clients in an IID manner, with 50% of clients randomly selected to participate in each round. Our code is available at https://github.com/15660733568/Yyk_FAL.

Adversarial robustness is evaluated using the $\ell_\infty$ PGD attack (Madry et al., 2018) with step size $\delta/5$, 10 attack iterations, and maximum perturbation $\delta$. Depending on the adversarial training scheme, model updates are performed using either first-order (SGD with momentum 0.5) or zeroth-order methods. Zeroth-order estimators based on random direction finite differences are used when gradients are unavailable, and the Moreau envelope is employed to smooth non-smooth objectives.

#### D.1.1. EVALUATION METRIC.

We report the generalization gap, defined as the absolute difference between adversarial training accuracy and adversarial test accuracy:

$$
\text{Generalization Gap} = |\text{AdvTrainAcc} - \text{AdvTestAcc}|.
$$

#### D.1.2. EXPERIMENTAL ANALYSIS

- **Impact on client participation ratio.** Increasing the client participation ratio incorporates a larger portion of the dataset in each communication round, thereby facilitating the training of more accurate global models. This leads to a reduction in the generalization gap and an improvement in robust accuracy, as illustrated in Figure 6.

- **Impact on total client numbers.** Given a fixed total data volume, increasing the number of participating clients decreases the amount of local data available per client. This reduction leads to a widening of the generalization gap and a deceleration in the improvement of adversarial accuracy, underscoring the necessity of maintaining sufficient local data per participant to achieve effective robust learning (Figure 3).

We also extended our experiments to other datasets. Experimental results on the SVHN, Webspam, and Epsilon datasets consistently show that increasing the attack strength $\delta$ leads to a significant drop in test accuracy and a corresponding widening of the generalization gap. Beyond reporting the empirical metrics from previous experiments (Table 4), we further evaluate our approach under a non-i.i.d. regime on the MNIST dataset (Table 5).

### D.2. Federated Adversarial Learning Algorithms.

**Federated Adversarial Learning.** FAL implements a federated adversarial training procedure where, at each communication round, a subset of clients is randomly selected. Each client initializes its local model with the current global model and performs local adversarial training by generating adversarial examples (e.g., via PGD) within its local minibatches. The clients compute local gradients using these adversarial examples and update their local models accordingly. Subsequently, local updates are uploaded to the server, which aggregates them via weighted averaging to update the global model. Although FAL achieves strong empirical robustness, its reliance on non-smooth adversarial objectives results in degradation of generalization performance as adversarial attack strength increases.

**FAME.** FAME enhances the federated adversarial learning framework by incorporating Moreau envelope smoothing to address the non-smoothness inherent in adversarial objectives. In each communication round, the server selects a subset of clients which locally perform adversarial training with smoothed objectives. Each client updates both its local model and a Moreau envelope parameter through iterative optimization steps. The Moreau envelope parameter acts as a smooth surrogate that approximates the original adversarial loss, enabling stability bounds independent of attack strength. Clients send both model and envelope updates to the server, which aggregates these updates to form the next global model and Moreau parameter, leading to improved adversarial robustness and generalization guarantees.

**FalZO.** FalZO addresses scenarios where gradient information is unavailable or unreliable, such as privacy-sensitive or black-box federated settings. It employs zeroth-order optimization by generating samples and direction vectors from local datasets and uniform distributions. Clients estimate gradients through function value evaluations combined with random perturbations, thus circumventing explicit gradient calculations. Local gradient estimates are aggregated at the server to update the global model iteratively. While zeroth-order methods generally yield weaker theoretical guarantees, FalZO demonstrates competitive robustness and generalization performance under challenging adversarial environments.

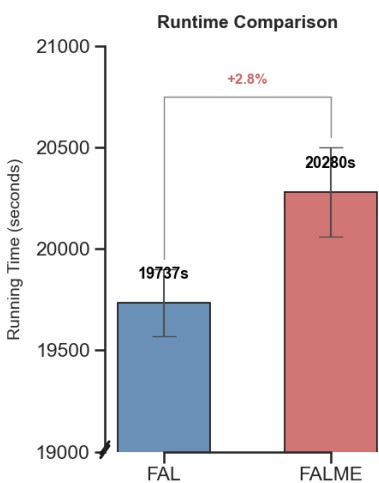

*Figure 5.* Average runtime (in seconds) of FAL and FALME.

*Table 4.* Robust Test Accuracy and Generalization Gap for Different Algorithms (On MNIST)

| Algorithm | $\delta = 0.1$ | | | $\delta = 0.3$ | | | $\delta = 0.5$ | | |
|---|---|---|---|---|---|---|---|---|---|
| | Train Acc | Test Acc | Gap | Train Acc | Test Acc | Gap | Train Acc | Test Acc | Gap |
| FAL | 97.97 | 94.77 | 0.032 | 97.85 | 93.15 | 0.047 | 97.71 | 91.30 | 0.064 |
| FalME | 98.74 | 95.72 | 0.030 | 98.57 | 94.36 | 0.042 | 98.41 | 93.20 | 0.052 |
| FalZO | 95.62 | 92.78 | 0.008 | 94.26 | 89.17 | 0.050 | 87.60 | 77.50 | 0.101 |

*Table 5.* Algorithm Evaluation under the Standard Non-IID Setting on the MNIST Dataset

| Algorithm | $L_\infty$ | PGD Num. | PGD Size | Train. Acc | Test. Acc | Gap |
|---|---|---|---|---|---|---|
| FAL | 0 | – | – | 94.0 | 75.0 | 0.190 |
| | 2/255 | 10 | 2/255 | 82.7 | 51.7 | 0.310 |
| | 4/255 | 10 | 2/255 | 82.8 | 45.0 | 0.378 |
| FalME | 0 | – | – | 94.2 | 74.8 | 0.194 |
| | 2/255 | 10 | 2/255 | 83.5 | 58.2 | 0.253 |
| | 4/255 | 10 | 4/255 | 80.4 | 48.2 | 0.322 |

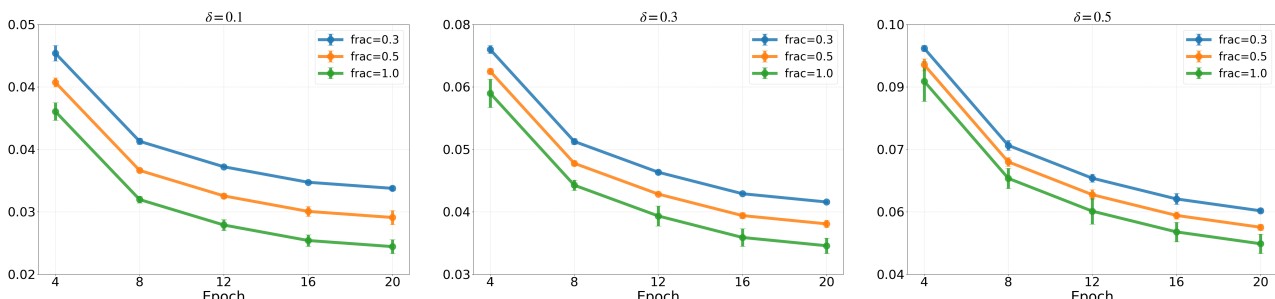

*Figure 6.* Results for generalization gap with different client participation ratio on MNIST

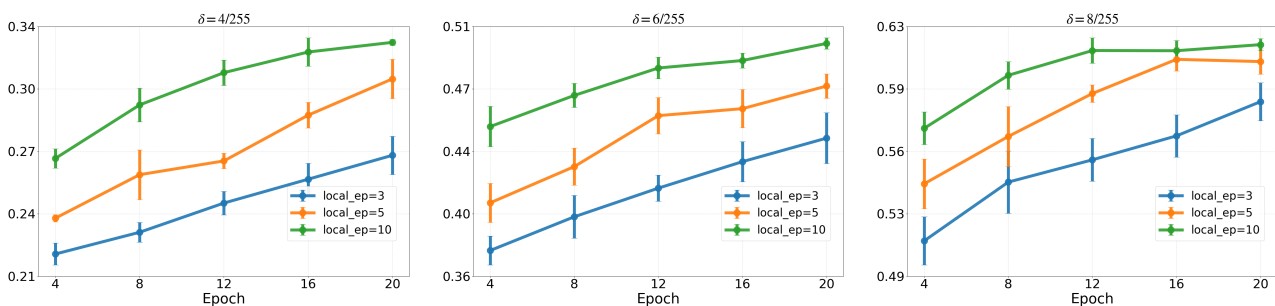

*Figure 7.* The generalization gap with different local training epochs on SVHN.

---

**Algorithm 1: Federated Adversarial Learning (FAL)**

---

**Input:** Initial global model $w^1$, total communication rounds $T$, number of selected clients per round $M$, local minibatch size $b_1$, local datasets $\{\mathcal{D}_i\}_{i=1}^N$
**Parameter:** Initial learning rate $\eta_1$, aggregation weights $\alpha_{i,j}^t$ (default 1).
**Output:** Final global model $w^T$

1: Let $t = 1$

2: **while** $t \leq T$ **do**

3:      Randomly select a set $M_t$ of $M$ clients

4:      Set learning rate $\eta_t = \eta_1/t$

5:      **for all** $i \in M_t$ **in parallel do**

6:           Set local model $w_i^t = w^t$

7:           **for** $j = 1$ **to** $b_1$ **do**

8:                Sample $z_{i,j}^t$ from $\mathcal{D}_i$

9:                Generate adversarial example $z_{i,j}^{t,adv}$ (e.g., PGD attack) using $w_i^t$

10:               Compute local gradient $\nabla f_i(w_i^t; z_{i,j}^{t,adv})$

11:           **end for**

12:           Compute local update

13:           $g_i^t = \frac{1}{b_1} \sum_{j=1}^{b_1} \nabla f_i(w_i^t; z_{i,j}^{t,adv})$

14:           Upload $g_i^t$ to the server

15:      **end for**

16:      **Server:** Update global model by

17:           $w^{t+1} = w^t - \eta_t \frac{1}{M} \sum_{i \in M_t} g_i^t$

18:      $t \leftarrow t + 1$

19: **end while**

20: **return** $w^T$

---

---

**Algorithm 2: Federated Adversarial Learning via Moreau Envelope (FalME)**

---

**Require:** Number of clients $N$, clients per round $M$, total rounds $T$; Adversarial budget $\delta$, stepsizes $\alpha_t \leq 1/p$ (or $\tau_t = 1 - \alpha_t p$), $\eta_t$; Loss functions $g_i(w; z_i)$, surrogate $f_i(w; z_i) = \max_{\|z_i - z_i'\|_p \leq \delta} g_i(w; z_i')$
$\qquad$ Initialize $w_i^0 = w^0$ for all $i \in [N]$

1:  Initialize: Global model $w^0$, local Moreau parameters $u_i^0 = w^0$ for all $i \in [N]$

2:  **for communication round** $t = 0$ **to** $T - 1$ **do**

3:  $\qquad$ Server selects subset $M_t$ of $M$ clients uniformly at random

4:  $\qquad$ Server broadcasts $w^t$ to all clients $i \in M_t$

5:  $\qquad$ **for each client** $i \in M_t$ **in parallel do**

6:  $\qquad\qquad$ **Local Adversarial Training:**

7:  $\qquad\qquad$ Initialize $w_{i,0}^t = w^t$

8:  $\qquad\qquad$ **for local step** $s = 0$ **to** $N - 1$ **do**

9:  $\qquad\qquad\qquad$ Compute adversarial example: $z_{i,s}^t = \arg\max_{\|z_i - z_i'\|_p \leq \delta} g_i(w_{i,s}^t; z_i')$

10: $\qquad\qquad\qquad$ Update local model: $w_{i,s+1}^{t+1} = w_{i,s}^{t+1} - \eta_t \nabla_w g_i(w_{i,s}^{t+1}, z_{i,s}^t)$

11: $\qquad\qquad$ **end for**

12: $\qquad\qquad$ Set $w_i^{t+1} = w_{i,N}^{t+1}$

13: $\qquad\qquad$ **Moreau Envelope Update:**

14: $\qquad\qquad$ Update local Moreau parameter: $u_i^{t+1} = u_i^t + \alpha_t p(w_i^{t+1} - u_i^t)$

15: $\qquad\qquad$ Send $w_i^{t+1}$ and $u_i^{t+1}$ to server

16: $\qquad$ **end for**

17: $\qquad$ **Server Aggregation:**

18: $\qquad$ Update global model: $w^{t+1} = \frac{1}{M} \sum_{i \in M_t} w_i^{t+1}$

19: $\qquad$ Update global Moreau parameter: $u^{t+1} = \frac{1}{M} \sum_{i \in M_t} u_i^{t+1}$

20: **end for**

**Ensure:** Global model $w^T$, Moreau parameter $u^T$

---

---

**Algorithm 3: Federated Adversarial Learning via Zeroth-Order Optimization (FalZO)**

---

**Require:** $w^1$: initial global model; $\eta_1$: initial learning rate; $b_1, b_2$: minibatch sizes for samples and direction vectors respectively; $\mu$: positive step size in the definition of the derivative; $M$: number of clients selected to update the global model in each iteration

1: **for all** $t = 1$ to $T - 1$ **do**

2:     Randomly select a clients set $M_t$, let $\eta_t = \eta_1/t$

3:     **for all** $i \in M_t$ **in parallel do**

4:        Let $w_i^t = w^t$

5:        Generate $\{z_{i,m}^t\}_{m=1}^{b_1}$ and $\{v_{il}^t\}_{l=1}^{b_2}$ from $\mathcal{D}_i$ and $d$-dimensional uniform distribution

6:        Compute $\sum_{m=1}^{b_1} \tilde{\nabla} f_i$ and upload it to global model

7:     **end for**

8:     Update $w^t$ to $w^{t+1}$

9: **end for**

**Ensure:** Final global model $w^T$

---

