# OpenReview forum: "Towards Understanding Generalization of Federated Adversarial Learning: Perspective of Algorithmic Stability"
_ICML.cc/2026/Conference — ICML 2026 regular_

### Official Review · Reviewer_D1Y1 · 2026-02-21

**Soundness:** 2
**Presentation:** 2
**Significance:** 1
**Originality:** 3
**Overall Recommendation:** 2
**Confidence:** 4

**Summary:**

This paper investigates the problem of generalization of federated adversarial learning (FAL) from the perspective of algorithmic stability. The paper first demonstrates that standard SGD-based FAL exhibits perturbation-dependent instability, where stronger adversarial attacks can lead to degraded generalization. To mitigate this issue, the paper leverages Moreau envelope optimization and proposes FalME, which smooths the non-smooth adversarial objective via proximal regularization, and enables stability bounds independent of
the attack strength. Furthermore, the analysis is extended to a zeroth-order variant (FalZO), enabling robustness guarantees in black-box federated environments where gradient information is unavailable.

**Compliance With Llm Reviewing Policy:**

Affirmed.

**Final Justification:**

I thank the authors for the additional results and analysis they provided. However, the comparison between FalME and FAL alone does not appear to be fully fair for assessing the performance gain of FalME, especially since other FAL methods that explicitly address heterogeneity were not included. Moreover, as shown by the authors, FalME suffers a substantial performance drop under heterogeneous data, decreasing from 74.8 to 48.2. This suggests a clear limitation in effectively handling the original task under realistic heterogeneous settings, which in turn significantly weakens the motivation and persuasiveness of the algorithmic stability analysis. I appreciate the additional theoretical discussion provided by the authors; however, the practical value of FalME remains substantially undermined by its weak performance on heterogeneous data. Therefore, I maintain my original score of 2 and my recommendation for rejection.

**Key Questions For Authors:**

1. The proof appears to rely on the assumption $w_i^t = w^t$. Does this correspond to a distributed-SGD regime (i.e., a single local step per round) or fully synchronous updates? How would the stability analysis change under the standard FedAvg setting with multiple local steps (K > 1), where client models naturally drift?

2. Do the theoretical stability bounds extend to the typical non-IID federated setting where client data distributions differ significantly and local updates amplify drift ($w_i^t \neq w^t$)?  If so, how is statistical heterogeneity reflected in the bound? If not, what additional assumptions would be required?

3. The FalME analysis claims $\delta$-independent stability. Are the Lipschitz and weak convexity constants used in the proof independent of the attack radius $\delta$, or could they implicitly scale with $\delta$ in adversarial settings? Clarifying this point would help assess the practical meaning of the perturbation-independent guarantee.

4. The experimental evaluation is primarily conducted under IID data partitions. Could the authors provide results under standard non-IID scenarios (e.g., Dirichlet-based label skew) to better assess robustness and generalization in realistic federated environments?

5. The formulation assumes equal local dataset sizes across clients. Do the theoretical guarantees extend to heterogeneous client sample sizes (i.e., varying $n_i$ across clients)?  If so, how would the stability bound change?

**Limitations:**

The theoretical analysis relies on several simplifying assumptions, including equal local dataset sizes and the absence of explicit modeling of multiple local steps in FedAvg. As a result, the stability bounds may not fully capture client drift or statistical heterogeneity commonly observed in realistic non-IID federated settings. In addition, the empirical evaluation is primarily conducted under IID data partitions, which limits the assessment of the proposed method in practical federated learning environments.

**Strengths And Weaknesses:**

Strengths：

1. The paper provides a theoretical framework for analyzing FAL generalization through the lens of algorithmic stability, which reveals a critical point of how adversarial perturbations influence generalization.

2. The paper introduces a Moreau-envelope formulation (FalME) that, under weak convexity, establishes uniform stability guarantees that do not depend on $\delta$, providing a theoretically grounded advance beyond conventional SGD-based FAL.

3. The framework is further extended to zeroth-order optimization, making it applicable to black-box federated scenarios.

4. The authors release code to support reproducibility.

Weaknesses:
1. The formulation assumes equal local sample sizes across clients (|S_i| = n), which simplifies the stability analysis. While this does not preclude distributional heterogeneity ($D_i$ may differ), it omits quantity skew commonly observed in real-world non-IID federated settings. It remains unclear whether the theoretical guarantees extend to heterogeneous client sample sizes.

2. The claim of $\delta$-independent stability in the Moreau-envelope variant may be less convincing, as the Lipschitz and weak convexity constants are not explicitly characterized with respect to the attack radius. If these constants implicitly depend on $\delta$, the practical significance of the perturbation-independent bound becomes unclear.

3. The notation in the problem formulation and assumptions is somewhat inconsistent. In particular, the index $i$ is used for both client indexing and data indexing (e.g., $z_i$ versus $z_{ij}$), which may introduce ambiguity in the stability analysis and weaken the clarity of the proofs.

4. In the proof of Theorem 4.1/4.3, the analysis assumes $w_i^t = w^t$, effectively eliminating client drift. This equality generally does not hold for FedAvg-style FL with multiple local steps/epochs, especially under non-IID data. Clarifying whether the theory targets a distributed-SGD regime (local steps K=1) or providing an extension accounting for local updates would strengthen the rigor and applicability.

5. In practical FL deployments, data across clients is typically non-IID. However, the experimental evaluation in this paper is primarily conducted under IID data partitions (e.g., MNIST), which does not reflect the core challenge of statistical heterogeneity in FL. As a result, it is unclear whether the proposed method would maintain its theoretical and empirical advantages under standard non-IID settings. The absence of experiments under standard non-IID scenarios (e.g., label skew or Dirichlet partitions) limits the practical relevance of the empirical validation.

6. While the experiments adopt a standard FedAvg-style setting with multiple local epochs per communication round, the theoretical analysis does not explicitly model or quantify the effect of local steps (K). In particular, the stability bounds do not depend on K, despite the well-known impact of local updates on client drift and instability under non-IID data. This mismatch creates a gap between the theoretical framework and the experimental setup.

---

> ### Author Rebuttal · Authors · 2026-03-30
>
> We thank the reviewer for the **rigorous and thoughtful evaluation**. Your comments regarding the scope of our stability analysis, especially concerning **heterogeneity and local updates**, are extremely valuable. We provide the following clarifications and theoretical extensions.
>
> Q1: On the assumption of equal local sample sizes.
>
> A1: We agree that the assumption $|S_i|=n$ simplifies exposition and does not capture quantity skew across clients. The analysis extends directly to **heterogeneous sample** sizes $n_i$, where the $1/n$ or $1/Nn$ terms are replaced by $n_i$-dependent weighted factors.
>
> Q2: On the meaning of δ-independence in FalME.
>
> A2: We thank the reviewer for this important question. More precisely, FalME **mitigates** rather than completely removes the effect of $\delta$. Its key benefit is not to make adversarial perturbations irrelevant, but to eliminate the explicit instability term caused by the **non-smooth max operator** in adversarial loss. In standard FAL, $\delta$ enters the stability bound explicitly through a term such as $L_z\delta$. Under FalME, this perturbation-induced non-smoothness is absorbed by the Moreau mechanism with $p>ℓ$, so it no longer appears explicitly in the final stability bound.
>
> Q3:  The current stability analysis under non-IID settings with multiple local steps and client drift.
>
> A3: We thank the reviewer for this important question. Our current theory is developed under a **synchronized-round** stability abstraction, which effectively corresponds to the $K=1$ regime. Accordingly, the proof uses $w_i^t=w^t$ to isolate the core stability effects of adversarial non-smoothness, Moreau smoothing, and ZO approximation, without yet explicitly parameterizing **local-drift amplification**.
>
> Under standard FedAvg training, **multiple local updates** ($K>1$) make local models drift away from the shared global iterate, and under **heterogeneous** data this drift appears as an additional $K$-dependent term in the stability recursion.
>
> (1)Our stability framework naturally extends to non-IID settings by incorporating a **local-drift condition** (following Ding et al., 2025):$\|\nabla F_i(w) - \nabla F(w)\| \leq C_{\delta} D_i$. where $D_{\max} = \max\limits_{i} D_i$. This assumption is natural because it directly quantifies the mismatch between each client objective and the global objective, and reduces to the IID case when $D_{\max} = 0$.
>
> (2) **Stability recursion**.Under the local-drift condition, the same stability recursion as in our proofs continues to hold, with heterogeneity entering as an additional additive term. Specifically, each local step contributes an **extra drift** of order $O(\eta_t D_i)$; **aggregating over K local steps** yields an additional $O(\eta_t D_{\max})$ term per round. Thus, the generic recursion becomes
> $\Delta_{t+1} \leq (1 + c_t) \Delta_t + \eta_t (E_t + K D_{\max})$,
>
> where $E_t$ is exactly the original term appearing in the IID analysis. In other words, neither non-IID heterogeneity nor $K>1$ local updates changes the core proof strategy; they only introduce a controlled drift term into the same one-step stability recursion.
>
> (3) **Unrolling the recursion**. Applying the same argument as in Theorems 4.3, 4.8, with the current **diminishing step-size** schedule yields the same $T^{1/2} \log T$ scaling, but with an extra additive heterogeneity term. This may leads to the following natural, K-aware refinements:
>
> FAL: $\mathcal{O} ( ( (Nn)^{-1} L + L_z \delta + K(\delta + 1) D_{\max} ) \sqrt{T \log T})$,
>
> FalME: $\mathcal{O} ( \frac{p}{p-\ell} ( (Nn)^{-1} L + n^{-1} + K D_{\max} ) \sqrt{T \log T})$,
>
> FalZO: $\mathcal{O} ( ( (Nn)^{-1} L + \mu + L_z \delta + K(\delta + 1) D_{\max} ) \sqrt{T \log T})$.
>
> Thus, FAL and FalZO retain $\delta$-dependent terms, whereas FalME remains $\delta$-independent due to Me smoothing, resulting in improved stability.
>
> Q4: The lack of standard non-IID experiments.
>
> A4: We have further evaluated our algorithms under standard **non-IID** settings, and the results (see Table 1) remain consistent with our theoretical predictions. In particular, while stronger heterogeneity enlarges the generalization gap across all methods, the relative ordering is preserved, with FalME remaining the most stable.
>
> **Table 1**
> |Algorithm|$L_\infty$|PGD Num.|PGD Size|Train. Acc|Test . Acc|Gap|
> |:-:|:-:|:-:|:-:|:-:|:-:|:-:|
> |FAL|0|--|--|94.0|75.0|0.190|
> || $2/255$|10|$2/255$ |82.7| 51.7|0.310|
> || $4/255$|10| $2/255$|82.8|45.0|0.378|
> |FalME|0|--|--|94.2|74.8|0.194|
> ||$2/255$|10|$2/255$|83.5|58.2|0.253|
> ||$4/255$|10| $4/255$|80.4|48.2|0.322|
>
> Q5: On notation ambiguity.
>
> A5: We thank the reviewer for pointing this out. In the revision, we will consistently separate client indices and sample indices, e.g., using $i$ for clients and $j$ for within-client samples, to avoid ambiguity such as $z_i$ versus $z_{ij}$.
>
> We hope these clarifications address the reviewer’s concerns and thank the reviewer for the constructive feedback.

---

> > ### Author Rebuttal · Reviewer_D1Y1 · 2026-04-02
> >
> > We thank the authors for their response, for providing additional experimental results, and for clarifying the technical terms in the paper. However, the significant performance drop observed under the non-IID setting suggests a clear limitation of the proposed theoretical prediction. This indicates that the effectiveness of the main contribution appears to rely largely on idealized settings, which substantially limits its practical applicability. In addition, from a theoretical perspective, it remains unclear how the proposed stability framework can be extended to non-IID scenarios. Given that heterogeneity and data imbalance are among the most fundamental characteristics of FL, this weakness substantially undermines the core motivation and significance of the paper. In our view, this concern cannot be fully addressed by the current method proposed in the paper.

---

> > > ### Author Response · Authors · 2026-04-03
> > >
> > > **Dear Reviewer**,
> > >
> > > Thank you for the careful follow-up and thoughtful comments. We agree that heterogeneity and data imbalance are fundamental in FL and should be addressed explicitly when assessing the scope of both the theory and the experiments. We also realize that our previous rebuttal did not make two points sufficiently explicit:
> > >
> > > (1)**Validity of the added non-IID experiment**.
> > >
> > > Our earlier added experiment was performed on **CIFAR-10 with ResNet under Dirichlet label skew**, a standard yet highly heterogeneous FL setting. The lower absolute accuracy in this case is expected rather than anomalous: even prior FAT results report $35.59$% adversarial test accuracy and $70.29$% original test accuracy on **federated CIFAR-10** (Table 2, Zizzo et al. (2020)). Hence, the key issue is not the absolute accuracy itself, but whether the relative advantage persists under stronger heterogeneity; our results show that it does, with FalME **still consistently outperforming FAL** (see Table 1).
> > >
> > > **Table 1 Training a CNN model on MNIST under non-IID data distributions**.
> > > |Algorithm|$L_\infty$|PGD Num.|PGD Size|Train. Acc|Test . Acc|Gap|
> > > |:-:|:-:|:-:|:-:|:-:|:-:|:-:|
> > > || 0|--|--|94.04|93.29|0.0075|
> > > |FAL|0.3|10|0.01|89.68|84.25|0.054|
> > > ||0.5|10|0.01|88.06|81.44|0.066|
> > > ||0|--|--|93.59|93.46|0.013|
> > > |FalME|0.3|10|0.01|89.91|86.19|0.037|
> > > ||0.5|10|0.01|89.48|84.38|0.051|
> > >
> > > **Table 2 Adversarial and original test accuracy of FL**.
> > > |Dataset|$L_\infty$|PGD Num.|PGD Size|Adv.test.acc|test.acc|
> > > |:-:|:-:|:-:|:-:|:-:|:-:|
> > > |CIFAR10|8/255|10|2/255|35.39|70.29
> > >
> > > Table 1 shows that this advantage persists under non-IID MNIST: compared with standard FAL, FalME **consistently improves adversarial test accuracy and reduces the generalization gap**, further supporting that the benefit of Moreau smoothing is not limited to IID settings.
> > >
> > > (2)**On how the stability framework extends to non-IID settings**.
> > >
> > > Proof sketch: The proof follows the same line of argument as that of Theorem 4.8. Let
> > >
> > > $M_i(u_i;S_i) :=\min_{w_i}\frac{1}{n}\sum_{j=1}^n(f_i(w_i;z_{ij})+\frac{p}{2}\|w_i-u_i\|^2), M(u;S):=\frac{1}{N}\sum_{i=1}^NM_i(u_i;S_i)$,
> > >
> > > and let
> > >
> > > $w_i^\star(u_i;S_i):=\arg\min_{w_i}\frac{1}{n}\sum_{j=1}^n(f_i(w_i;z_{ij})+\frac{p}{2}\|w_i-u_i\|^2)$.
> > >
> > > Then
> > >
> > > $\nabla M_i(u_i;S_i)=p(u_i-w_i^\star(u_i;S_i)),$
> > >
> > > and, as in the proof of Theorem 4.8,
> > >
> > > $\| \nabla M_i(u_i; S_i) - \nabla M_i(v_i; S_i) \| \leq L_w \| u_i - v_i \|,  L_w = \max ( p, \frac{p\ell}{p - \ell})$.
> > >
> > > Assume the heterogeneity condition (Ding et al., 2025)
> > >
> > > $\|\nabla M_i(u_i;S)-\nabla M(u;S)\|\le C_H D_i, D_{\max}:=\max_{i\in[N]}D_i$.
> > >
> > > For local step $k=0,\dots,K-1$, define
> > >
> > > $u_{i,0}^t=u_t,u_{i,k+1}^t=u_{i,k}^t-\alpha_t \nabla M_i(u_{i,k}^t;S_i)$, and the server aggregation $u_{t+1}=\frac{1}{M}\sum_{i\in\mathcal M_t}u_{i,K}^t $.
> > >
> > > Now couple two neighboring datasets $S,S'$ and define
> > >
> > > $\Delta_{t,k}:=\mathbb E\|u_{i,k}^t(S)-u_{i,k}^t(S')\|, \Delta_t:=\mathbb E\|u_t(S)-u_t(S')\|$.
> > >
> > > Then
> > >
> > > \begin{align*}\Delta_{t,k+1}&=\mathbb E\|u_{i,k}^t(S)-u_{i,k}^t(S')-\alpha_t(\nabla M_i(u_{i,k}^t(S);S_i)-\nabla M_i(u_{i,k}^t(S');S_i'))\| \\ \le(1+\alpha_tL_w)\Delta_{t,k}+\alpha_t \mathcal E_t+\alpha_t C_H D_i,\end{align*}
> > >
> > > where $\mathcal E_t$ is the same replacement term as in the IID FalME proof, and
> > > $\mathcal E_t \le C\,\frac{p}{p-\ell}(\frac{L}{Nn}+\frac{1}{n})$.
> > >
> > > Iterating over $K$ local steps yields
> > >
> > > $\Delta_{t,K}\le(1+c\alpha_tK)\Delta_t+c\alpha_tK[\frac{p}{p-\ell}(\frac{L}{Nn}+\frac{1}{n})+D_{\max}]$.
> > >
> > > After aggregation, we obtain
> > >
> > > $\Delta_{t+1} \le (1+c\alpha_tK) \Delta_t+c\alpha_t[\frac{p}{p-\ell}(\frac{L}{Nn}+\frac{1}{n})+K D_{\max}]$.
> > >
> > > Choose the diminishing step-size $\alpha_t=\frac{\alpha_1}{t}, \alpha_1<\frac{1}{L_w}$. Then
> > >
> > > $\Delta_{t+1}\le(1+\frac{c_1}{t})\Delta_t+\frac{c_2}{t}[\frac{p}{p-\ell}(\frac{L}{Nn}+\frac{1}{n})+K D_{\max}]$.
> > >
> > > Unrolling the recursion gives
> > >
> > > $\Delta_T\le C[\frac{p}{p-\ell}(\frac{L}{Nn}+\frac{1}{n})+K D_{\max}]\sqrt{T}\log T$.
> > >
> > > Finally, by the uniform-stability-to-generalization argument used in Theorem 4.8, $\epsilon_{\mathrm{gen}} \le L\Delta_T$. Absorbing constants into the prefactor, we obtain the compact form
> > >
> > > $\epsilon_{\mathrm{gen}}\le O(\frac{p}{p-\ell}(\frac{L}{Nn}+\frac{1}{n}+K D_{\max})\sqrt{T}\log T)$.
> > >
> > > Therefore, heterogeneity and multiple local steps introduce the **additional controlled drift term $KD_{\max}$** into the same FalME stability recursion, while the bound still contains no explicit $\delta$-dependent instability term. The analyses of the other two algorithms proceed analogously, and the corresponding results can be obtained by the same line of argument.
> > >
> > >
> > > We **sincerely thank** the reviewer again for the thoughtful comments and respectfully hope these clarifications will be taken into consideration in the final assessment.
> > >
> > > Reference
> > >
> > > [1]Ding, W., An, Y. 2025. How does the smoothness approximation method facilitate generalization for federated adversarial learning?
> > >
> > > [2]Zizzo, G.. 2020. Fat: Federated adversarial training.

---

### Official Review · Reviewer_aWvh · 2026-02-28

**Soundness:** 3
**Presentation:** 3
**Significance:** 4
**Originality:** 4
**Overall Recommendation:** 5
**Confidence:** 4

**Summary:**

This paper systematically analyzes the generalization behavior of federated adversarial learning from the perspective of algorithmic stability. It reveals the negative impact of attack strength on stability and proposes the first $\delta$-independent generalization bound through Moreau-envelope smoothing. Furthermore, the analysis is extended to black-box settings via zeroth-order optimization.

**Compliance With Llm Reviewing Policy:**

Affirmed.

**Final Justification:**

N/A

**Key Questions For Authors:**

### Questions:
1. In general, theoretical guarantees for zeroth-order optimization methods typically exhibit explicit dependence on the parameter dimension d. However, the final generalization bound presented in the paper does not clearly reflect such dimensional dependence. It appears that around Line 1055, a relaxation or bounding step is applied that effectively absorbs or suppresses the d-related terms. It is unclear whether this simplification is fully justified or whether it may conceal an inherent dimensional dependency.
2. Does the theoretical analysis rely on specific assumptions about the client data distributions? In particular, would the proposed stability bounds still hold under more severely non-IID and highly heterogeneous federated settings, where client data distributions differ significantly?

**Limitations:**

See weakness and questions

**Strengths And Weaknesses:**

### Strengths:
1. This paper constructs a unified and systematic theoretical framework for algorithm stability, which covers both first-order and zero-order federated adversarial optimization methods from the same analytical perspective. The theoretical structure is clear and has strong integrity.
2. By introducing the Moreau-envelope smoothing technique, this paper derives for the first time an upper bound on the generalization error that does not depend on the attack strength $\delta$ in federated adversarial learning, theoretically alleviating the problem of generalization degradation caused by strong adversarial perturbations.
3. This paper further extends the theoretical analysis to black-box optimization scenarios, ensuring the feasibility and stability of the method in privacy-constrained federated environments where gradients are unavailable, thus enhancing its practical application value.

### Weaknesses:
1. The theoretical performance of FalME depends on the choice of the smoothing parameter $p>\ell$; however, the paper lacks systematic guidance on how to select this parameter in practical training scenarios and does not provide a sensitivity analysis to illustrate its empirical impact.
2. Although the paper establishes a clear stability bound that is independent of the attack strength $\delta$, it is intuitively natural for the generalization error to depend on $\delta$ in adversarial learning. From this perspective, the role of the Moreau envelope smoothing technique seems more aligned with mitigating the dependence on $\delta$, rather than completely eliminating it, which is also consistent with the experimental results. Furthermore, the paper does not theoretically characterize the relationship between the smoothing parameter in the Moreau envelope and the attack strength $\delta$.
3. Although the paper primarily focuses on theoretical contributions, the experimental evaluation lacks comparisons with other existing generalization-enhancing methods in federated or adversarial learning.

---

> ### Author Rebuttal · Authors · 2026-03-30
>
> We sincerely thank the reviewer for the positive assessment and for recognizing the **originality, significance, and unified theoretical framework** of our work. We also appreciate the constructive suggestions, which will help further strengthen the paper.
>
> Q1: On the choice of the Moreau smoothing parameter $p>ℓ$.
>
> A1: We thank the reviewer for this important question. In our experiments, we set the Moreau smoothing parameter to $p=0.0005$. We also evaluated several nearby choices during tuning and found that the empirical performance varied only marginally, as reported in Table 1 below, indicating that FalME is **not highly sensitive** to the precise value of $p$.
>
> **Theoretically**, the key requirement is simply $p>ℓ$, which ensures **strong convexity** of the inner problem. Taken together, these results suggest that p can be chosen in a stable and practically robust manner, without delicate parameter tuning. We will clarify this point and include the sensitivity evidence in the revision.
>
> **Table 1**
> |Algorithm|$L_\infty$|PGD Num.|PGD Size|p|Train. Acc|Test . Acc|Gap|
> |:-:|:-:|:-:|:-:|:-:|:-:|:-:|:-:|
> |FalME|$2/255$|10|$2/255$|0.0005|83.53|58.20|0.253|
> |FalME|$2/255$|10|$2/255$|0.0008|83.60|57.90|0.257|
>
> Q2: On whether δ-dependence is eliminated or mitigated.
>
> A2: We thank the reviewer for this important question. More precisely, FalME **mitigates** rather than completely removes the effect of $\delta$. Its key benefit is not to make adversarial perturbations irrelevant, but to eliminate the explicit instability term caused by the **non-smooth max operator** in adversarial loss. In standard FAL, δ enters the stability bound explicitly through a term such as $L_z\delta$. Under FalME, this perturbation-induced non-smoothness is absorbed by the **Moreau mechanism** with p>ℓ, so it no longer appears explicitly in the final stability bound.
>
> Q3: Lack of comparison with existing generalization methods.
>
> A3: We appreciate this suggestion. Our experiments are designed to isolate and **validate** the core **theoretical mechanisms** under a unified FAL framework, avoiding confounding factors from heterogeneous designs. Thus, the focus is on theory verification rather than broad benchmarking.
>
> Q4: On dimensional dependence in FalZO.
>
> A4: We thank the reviewer for this important observation. We agree that the current final FalZO bound hides the dimensional dependence. In the revision, we will keep the d-dependent terms explicit.
>
> Q5: On client distribution assumptions and heterogeneity.
>
> A5: We thank the reviewer for this important question.
>
> （1）Our stability framework naturally extends to non-IID settings by incorporating a **local-drift condition** (e.g., following Ding et al., 2025):$\|\nabla F_i(w) - \nabla F(w)\| \leq C_{\delta} D_i$. where $D_{\text{max}} = \max\limits_{i} D_i$. This assumption is natural because it directly quantifies the mismatch between each client objective and the global objective, and reduces to the IID case when $D_{\max}$ = 0.
>
> （2）**Stability recursion**. Under the local-drift condition, the same stability recursion as in our proofs continues to hold, with heterogeneity entering as an additional additive term. Specifically, each local step contributes an **extra drift** of order $\mathcal{O}(\eta_t D_i)$; **aggregating over $K$ local steps** yields an additional $\mathcal{O}(\eta_t D_{\max})$ term per round. Thus, the generic recursion becomes
>
> $\Delta_{t+1} \leq (1 + c_t) \Delta_t + \eta_t ( E_t + K D_{\text{max}} )$
>
>
> where $E_t$  is exactly the original term appearing in the IID analysis. In other words, non-IID heterogeneity does not alter the proof structure; it only adds a controlled drift term to the same one-step stability recursion.
>
> （3）**Unrolling the recursion**. Applying the same argument as in Theorems 4.3, 4.8, and 4.11 with the current **diminishing step-size** schedule yields the same $T^{1/2} \log T$ scaling, but with an extra additive heterogeneity term. **Hnece**, we may roughly infer the following non-IID variants:
>
> FAL: $\mathcal{O} \left( \left( (Nn)^{-1} L + L_z \delta + K(\delta + 1) D_{\text{max}} \right) \sqrt{T \log T} \right)$,
>
> FalME: $\mathcal{O} \left( \frac{p}{p-\ell} \left( (Nn)^{-1} L + n^{-1} + K D_{\text{max}} \right) \sqrt{T \log T} \right)$,
>
> FalZO: $\mathcal{O} \left( \left( (Nn)^{-1} L + \mu + L_z \delta + K(\delta + 1) D_{\text{max}} \right) \sqrt{T \log T} \right)$.
>
> Thus, heterogeneity enlarges the generalization gap for all methods, but the relative ordering remains unchanged: FAL and FalZO retain $\delta$-dependent terms, whereas FalME remainsδ-independent due to Moreau smoothing, resulting in **improved stability**.
>
> Overall, we appreciate the reviewer’s strong support and insightful comments. We will revise the paper to make the role of $p$, the asymptotic treatment of dimensional factors in FalZO, and the scope of the non-IID setting more explicit.

---

> > ### Author Rebuttal · Reviewer_aWvh · 2026-04-03
> >
> > The authors have provided a thorough and well-organized rebuttal that addresses the majority of my key concerns. This paper makes a significant theoretical contribution for federated adversarial learning. The theoretical insights are valuable to the community. I maintain my positive assessment.

---

> > > ### Author Response · Authors · 2026-04-04
> > >
> > > We sincerely thank the reviewer for the positive evaluation of our work. We are truly grateful for the recognition of our response.

---

### Official Review · Reviewer_mZcj · 2026-03-09

**Soundness:** 2
**Presentation:** 3
**Significance:** 4
**Originality:** 4
**Overall Recommendation:** 5
**Confidence:** 3

**Summary:**

This paper analyzes generalization for federated adversarial learning (FAL) via uniform stability.  Their bounds demonstrate that FAL leads to an error floor related to the size of the adversarial perturbation and propose FalME algorithm which uses Moreau envelope smoothing and removes this dependency.  They also show that their framework for analysis can be extended to zeroth order optimization with FAL objective (FalZO).  The authors present experiments across image datasets and scale vector datasets comparing these different algorithms and demonstrating the that trends in empirical impact of the adversarial perturbation size aligns with their theoretical results.

**Compliance With Llm Reviewing Policy:**

Affirmed.

**Final Justification:**

I believe this is a strong theoretical paper analyzing federated adversarial learning.  In my initial review, my comments were mainly minor and the main point of deduction was the presentation of Table 1 summary of prior work where the T scaling seemed off for non-federated learning.  This has been addressed in the author's rebuttal along with providing some additional experimental results to address some other more minor feedback I had.

**Key Questions For Authors:**

See Suggestions in Strengths and Weaknesses

**Limitations:**

yes

**Strengths And Weaknesses:**

Strengths:
- Clearly written and well-structured.  The organization of the paper was easy to follow, I liked how each section of theoretical analysis included an introduction for the algorithm that is analyzed and each theorem is followed by a remark interpreting the derived bounds.
- Studies an area where there is not much prior work (generalization in federated adversarial learning) and proposes stronger bounds on generalization via uniforms stability while previous related work looks at on-average stability.  Additionally, compared to prior work their bounds seem to be better with scaling in terms of number of updates T and number of samples.
- Derives uniforms stability bounds on FAL, FalME, and FalZO.  FalME is proposed by the authors to remove the error floor induced by the size of the adversarial perturbation.  I thought this was quite significant and novel.  Theoretical results also have reasonable assumptions and proofs are provided (I did not thoroughly check the proofs though)
- Experimental results on multiple datasets to demonstrate trends in generalization gap align with those expected via the theoretical analysis.

Suggestions:
- Presentation:
    - In remark 4.9, it would be good to also note the added $\frac{1}{n}$ term in the FalME bound which suggests that not just the total number of samples matters for FalME, but also the number of samples per client.
    - Table 1 gives a nice summary of results and comparison to prior work, but it would be nice to also have some more discussion in the text which discusses these comparisons in more depth (ie. how comparable are these results, any other assumptions made in these papers, etc.)  Given all these prior work also seem to all have an error floor, this discussion can also more strongly highlight the contribution of FalME.
        - One comment I have about Table 1 is that the presented bound for Xiao et al. 2022 assumes a constant step size of $\frac{1}{L_w}$.  The result from the Xiao et al. 2022 paper is more general though and written with a $\sum_{t=1}^T \alpha_t$ term where $\alpha_t$ is the step size that satisfies $\alpha_t \le \frac{1}{L_w}$.  I think that given that this work uses a diminishing step size in the derived bounds, it would make sense to also present the Xiao et al. bound with diminishing step size of $\alpha_t = \frac{1}{L_w t}$.  This would lead to $\log T$ instead of $T$ scaling.  I think this makes more sense too since I would expect federated learning to scale worse with $T$ than the standard setting.
- Experiments:
   - Generalization gap vs delta- it would be nice to see a plot of the generalization gap over delta for FAL, FalME, and FalZO.  I'm picturing something similar to the plots in Figure 1 where you have 1 line for each algorithm, but each algorithm is run for the same fixed number of epochs and the x axis is delta.  Ideally, if aligned well with theory, we would expect to see that FalME's generalization gap stays relatively constant as delta increases while FAL and FalZO both scale with delta.
   - Achieved test accuracy/loss values for each algorithm - Generalization gap can decrease either due to improved test performance or increased training loss, so from a more practical standpoint it would be nice to see the actual test values.  Ideally FalME shouldn't be making it very hard to actually fit the training data well, thus leading to better generalization on the test data without actually any improvement on the test performance.
- Minor:
    - Some fragments in text:
         - Theorem 4.3: "For the FAL algorithm A."
         - Line 225 in Col 1 (Within remark 4.4): "Unlike the standard generalization error term which scales with the inverse sample size $(nN)^{-1}$."

---

> ### Author Rebuttal · Authors · 2026-03-30
>
> We sincerely thank the reviewer for the encouraging and constructive feedback, and for recognizing the **significance and originality** of our work. Below, we address the raised concerns through additional experiments and theoretical clarifications.
>
> Q1: On clarifying and expanding the comparison in Table 1.
>
> A1: We appreciate the suggestion and will **expand the discussion** around Table 1 to more **clearly distinguish** directly comparable results from those obtained under different heterogeneity assumptions and step-size regimes.
>
> We agree that the bound of Xiao et al. (2022) is stated in the more general form t=1Tαt. Accordingly, under the **diminishing step size** $\alpha_t = 1/(L_wt)$, the convex-case dependence should scale as $\log T$ rather than $T$, whereas their **non-convex** result remains linear in $T$. In our analysis, the $\log T$ factor arises from **unrolling the stability recursion** with $ηt≍1/t$ and applying the **harmonic-sum** bound $\sum_{t=1}^{T} \frac{1}{t} \leq \log (eT)$, which, together with the recursion multiplier, yields the stated  $\mathcal{O}(T^{1/2}\log T)$  bound under $\eta_1\le1/(2L_w)$.
>
> Q2: The generalization gap versus $\delta$.
>
> A2: Following your advice, we give the generalization gap against $\delta\in[0.1,0.5]$. As shown in Table 1 below, for standard FAL, the gap increases with $\delta$. In contrast, FalME's gap remains **better stable**, indicating that it effectively **mitigates** the impact of $\delta$ on the generalization gap. We will include this figure of the final version.
>
> **Table 1 Training a CNN model on MNIST under non-IID data distributions** .
> | Algorithm|$\delta$ | Train Acc | Test Acc |Gap |
> |:-:|:-:|:-:|:-:|:-:|
> |FAL|0.1|97.97|94.77|0.032|
> ||0.3|97.85|93.15|0.047|
> ||0.5|97.71|91.30|0.064|
> |FalME|0.1|98.74|95.72|0.030|
> ||0.3|98.57|94.36|0.042|
> ||0.5|98.41|93.20|0.052|
> |FalZO|0.1|95.62|92.78|0.008|
> ||0.3| 94.26|89.17|0.050|
> ||0.5|87.60|77.50|0.101|
>
> Q3: On reporting achieved test accuracy.
>
> A3: We agree that raw adversarial metrics are essential for practical interpretability. Our additional results (see Table 1) show that FalME does not merely reduce the gap by "making training harder"; it consistently **improves adversarial test accuracy** over standard FAL. This suggests that Moreau smoothing helps the model **generalize better** to unseen adversarial samples without sacrificing training convergence.
>
> Q4: The additional $1/n$ term in Remark 4.9.
>
> A4: We will add the discussion of the $1/Nn$ term in Remark 4.9 to emphasize that FalME benefits from both the number of clients ($N$) and samples per client ($n$).
>
> Q5: On the minor textual issues.
>
> A5: Thank you for catching these presentation issues. We will correct the noted text fragments and carefully proofread the final version.
>
> Overall, we are grateful for the reviewer’s supportive and technically informed feedback. We will incorporate these suggestions in the revision to make the paper clearer and more complete.

---

> > ### Author Rebuttal · Reviewer_mZcj · 2026-04-01
> >
> > Thank you for your response, I believe all my concerns were addressed and will raise my score to a 5.

---

> > > ### Author Response · Authors · 2026-04-03
> > >
> > > We sincerely thank the reviewer for the careful consideration and positive evaluation of our work. We are truly grateful for the recognition of our response and revisions.

---

### Official Review · Reviewer_y3fn · 2026-03-10

**Soundness:** 3
**Presentation:** 3
**Significance:** 3
**Originality:** 3
**Overall Recommendation:** 3
**Confidence:** 3

**Summary:**

This paper presents a theoretical analysis of generalization in Federated Adversarial Learning using algorithmic stability, introducing and evaluating three distinct algorithms: a standard SGD-based approach (FAL) whose generalization bound is dependent on attack strength δ; a Moreau-envelope smoothed variant that achieves δ-independent stability, thereby improving robustness; and a zeroth-order method designed for black-box, privacy-sensitive settings. The authors validate their theoretical findings with experiments showing that Moreau-envelope smoothing significantly reduces the robust generalization gap and enhances adversarial accuracy compared to standard FAL.

**Compliance With Llm Reviewing Policy:**

Affirmed.

**Final Justification:**

After reading the paper and the rebuttal carefully, I find the theoretical contribution solid and the authors’ responses convincing. In my original review, I raised several concerns regarding the scalability to larger datasets, the practical trade‑offs of FalZO, the behavior under non‑IID data, and the clarity of the appendix. The rebuttal added non‑IID experiments on CIFAR‑10 with ResNet, clarified the extension to heterogeneous settings via a local‑drift condition, and addressed concerns about FalZO’s trade‑offs. This is a solid, well-organised theoretical paper with clear contributions.

**Key Questions For Authors:**

（1）Have you tested FalME or FalZO on more complex datasets with larger neural network architectures？If not, do you anticipate any scalability challenges when applying these methods to more realistic federated settings?
（2）Could you comment on the computational cost and convergence speed of FalZO compared to first-order methods like FAL and FalME?Understanding the practical trade-offs would be valuable for practitioners considering its use in real-world, gradient-free environments.
（3）How do you expect the generalization bounds and empirical performance of the three algorithms to change under non-IID data distributions across clients?

**Limitations:**

Include a discussion of potential negative societal impacts, such as the dual-use nature of adversarial robustness research or privacy implications in federated settings

**Strengths And Weaknesses:**

This is a strong, high-quality paper that makes a significant and original contribution to the field. Its theoretical analysis is sound, its presentation is clear, and its findings have practical implications for designing more robust and generalizable federated learning systems. The combination of rigorous theory, novel algorithmic contributions, and supporting experiments makes it a valuable piece of research. However, the paper does have some limitations. First, the empirical validation is conducted primarily on relatively small-scale datasets; Experiments on more complex datasets would further substantiate the practical relevance of the theoretical claims. Second, while the appendix contains detailed proofs, its presentation is somewhat cluttered, which may hinder readability for researchers seeking to verify or build upon the derivations.

---

> ### Author Rebuttal · Authors · 2026-03-30
>
> We sincerely thank the reviewer for the constructive feedback and for recognizing the **soundness, originality, and practical relevance** of our work. We have addressed your concerns through additional experiments and clarified the theoretical extensions below.
>
> Q1: On empirical scale and scalability to larger datasets and architectures.
>
> A1: We appreciate the suggestion. While the initial submission focused on MNIST and SVHN, we have since extended the evaluation to CIFAR-10 with ResNet under non-IID settings. And we plotted the robust generalization gap against $\delta \in [0, 2/225]$. As shown in Table $1$ below, for standard FAL the gap increases with $\delta$ (from $0.195$ to $0.310$), while for FalME it **stays smaller** and **more stable** (from $0.194$ to $0.25$). This directly supports Theorem 4.8.
>
> **Table 1** Training a ResNet model on CIFAR-10 under non-IID data distributions.
> |Algorithm|$L_\infty$|PGD Num.|PGD Size|Train. Acc|Test . Acc|Gap|
> |:-:|:-:|:-:|:-:|:-:|:-:|:-:|
> || 0|--|--|94.0|75.0|0.190|
> |FAL| $2/255$|10|$2/255$ |82.7| 51.7|0.310|
> || $4/255$|10| $2/255$|82.8|45.0|0.378|
> ||0|--|--|94.2|74.8|0.194|
> |FalME|$2/255$|10|$2/255$|83.5|58.2|0.253|
> ||$4/255$|10| $4/255$|80.4|48.2|0.322|
>
> Q2: On the cost and convergence of FalZO.
>
> A2: This is a crucial practical consideration. Zeroth-order methods are generally more expensive and slower to converge than first-order methods. However, FalZO is intended for black-box or privacy-sensitive federated settings where gradients are inaccessible, and thus serves as a necessary **gradient-free** alternative rather than a replacement for standard FAL. We will clarify this trade-off in the revision through a concise comparison of runtime and iteration complexity.
>
> Q3: On the expected behavior under non-IID client distributions.
>
> A3: We thank the reviewer for this important question.
>
> （1）Our stability framework naturally extends to non-IID settings by incorporating a **local-drift condition** (e.g., following Ding et al., 2025):$\|\nabla F_i(w) - \nabla F(w)\| \leq C_{\delta} D_i$. where $D_{\text{max}} = \max\limits_{i} D_i$. This assumption is natural because it directly quantifies the mismatch between each client objective and the global objective, and reduces to the IID case when $D_{\max}$ = 0.
>
> （2）**Stability recursion**. Under the local-drift condition, the same stability recursion as in our proofs continues to hold, with heterogeneity entering as an additional additive term. Specifically, each local step contributes an **extra drift** of order $\mathcal{O}(\eta_t D_i)$; **aggregating over $K$ local steps** yields an additional $\mathcal{O}(\eta_t D_{\max})$ term per round. Thus, the generic recursion becomes
>
> $\Delta_{t+1} \leq (1 + c_t) \Delta_t + \eta_t ( E_t + K D_{\text{max}} )$
>
>
> where $E_t$  is exactly the original term appearing in the IID analysis. In other words, non-IID heterogeneity does not alter the proof structure; it only adds a controlled drift term to the same one-step stability recursion.
>
> （3）**Unrolling the recursion**. Applying the same argument as in Theorems 4.3, 4.8, and 4.11 with the current **diminishing step-size** schedule yields the same $T^{1/2} \log T$ scaling, but with an extra additive heterogeneity term. **Hnece**, we may roughly infer the following non-IID variants:
>
> FAL: $\mathcal{O} \left( \left( (Nn)^{-1} L + L_z \delta + K(\delta + 1) D_{\text{max}} \right) \sqrt{T \log T} \right)$,
>
> FalME: $\mathcal{O} \left( \frac{p}{p-\ell} \left( (Nn)^{-1} L + n^{-1} + K D_{\text{max}} \right) \sqrt{T \log T} \right)$,
>
> FalZO: $\mathcal{O} \left( \left( (Nn)^{-1} L + \mu + L_z \delta + K(\delta + 1) D_{\text{max}} \right) \sqrt{T \log T} \right)$.
>
> Thus, heterogeneity enlarges the generalization gap for all methods, but the relative ordering remains unchanged: FAL and FalZO retain $\delta$-dependent terms, whereas FalME remainsδ-independent due to **Moreau smoothing**, resulting in **improved stability**.
>
> Q4: The request to discuss broader societal impacts.
>
> A4: Adversarial robustness research is inherently dual-use, as it can both strengthen defenses and inform stronger attacks, while federated settings may still entail privacy risks despite avoiding raw-data sharing.
>
> Q5: On appendix clarity.
>
> A5: We appreciate this constructive suggestion. In the revision, we will restructure it into four parts: (A) notation and auxiliary lemmas, (B) proofs of the main results grouped by algorithm, (C) algorithmic details, and (D) experimental details and additional results.
>
> We hope these clarifications address the reviewer’s concerns and thank the reviewer for the constructive feedback.

---

> > ### Author Rebuttal · Reviewer_y3fn · 2026-04-06
> >
> > The authors supplemented rigorous non-IID experiments on CIFAR-10 with ResNet to verify their methods’ scalability on complex datasets and larger architectures. The supplementary theoretical and empirical results strengthen the paper’s core contributions, completeness, and practical value

---

> > > ### Author Response · Authors · 2026-04-06
> > >
> > > Dear Reviewer,
> > >
> > > We **sincerely thank you** for confirming that our responses have resolved the concerns raised in your initial review.
> > >
> > > As detailed in our previous responses, we extended the empirical evaluation beyond the original MNIST and SVHN settings to **a more challenging non-IID experiment** on CIFAR-10 using ResNet, and **clarified how our stability framework naturally extends to heterogeneous client distributions**. We also committed to improving the final version by refining the discussion of broader societal impacts and reorganizing the appendix for clarity.
> > >
> > > Given your confirmation that the concerns raised in your initial review **have been resolved**, we would be truly grateful to know whether any broader remaining reservations still stand in the way of a more favorable rating. If there are any additional questions or concerns that we have not yet addressed, we would be more than **happy to clarify** them.
> > >
> > > If our responses have indeed satisfactorily addressed the weaknesses identified in your initial review, **we respectfully hope you might generously consider reflecting this positive resolution in the final assessment**. **Thank you again** for your thoughtful and constructive feedback.

---

### Decision · Program_Chairs · 2026-04-30

**Decision:**

Accept (regular)

**Comment:**

The paper analyzes adversarial robustness in federated learning and proposes, and analysis, a method for improving generalization in the face of adversarial perturbation.  The contribution is substantiated through theoretical analysis, with supporting empirics.  The paper received mixed reviews, with three positive reviews supporting acceptance (even though one of them indicated 'weak reject'), and one more skeptical review (D1Y1) highlighting limitations and expressing valid concerns about relevance and some assumptions.